# LEARNING TEMPORAL CAUSAL REPRESENTATION UNDER NON-INVERTIBLE GENERATION PROCESS

## ABSTRACT

Identifying the underlying time-delayed latent causal processes in sequential data is vital for grasping temporal dynamics and making downstream reasoning. While some recent methods can robustly identify these latent causal variables, they rely on strict assumptions about the invertible generation process from latent variables to observed data. These assumptions are often hard to satisfy in real-world applications containing information loss. For instance, the visual perception process translates a 3D space into 2D images, or the phenomenon of persistence of vision incorporates historical data into current perceptions. To address this challenge, we establish an identifiability theory that allows for the recovery of independent latent components even when they come from a nonlinear and non-invertible mix. Using this theory as a foundation, we propose a principled approach, `CaRiNG`, to learn the **Ca**usal **R**epresentat**i**on of **N**on-invertible **G**enerative temporal data with identifiability guarantees. Specifically, we utilize the temporal context to recover lost latent information and employ the conditions in our theory to guide the training process. Through experiments conducted on synthetic datasets, we validate that the causal process is reliably identified by `CaRiNG`, even when the generation process is non-invertible. Moreover, we show that our approach considerably improves temporal understanding and reasoning in practical applications.

## 1 INTRODUCTION

Sequential data, including video, stock, and climate observations, are integral to our daily lives. Gaining an understanding of the causal dynamics in such time series data has always been a crucial challenge (Berzuini et al., 2012; Ghysels et al., 2016; Friston, 2009) and has attracted considerable attention. The core of this task is to identify the underlying causal dynamics in the data we observe.

Towards this goal, we focus on Independent Component Analysis (ICA) (Hyvärinen & Oja, 2000), which is a classical method for decomposing the latent signals from mixed observation. Recent advancements in nonlinear ICA (Hyvarinen & Morioka, 2016; 2017; Hyvarinen et al., 2019; Khemakhem et al., 2020; Sorrenson et al., 2020; Hälvä & Hyvarinen, 2020) have yielded robust theoretical evidence for the identifiability of latent variables, and enable the use of deep neural networks to address complex scenarios. For example, by assuming the latent variables in the data generation process are mutually independent, and leveraging the auxiliary side information such as time index, domain index, or class label, Hyvarinen & Morioka (2017); Hyvarinen et al. (2019); Hälvä & Hyvarinen (2020) have demonstrated the strong identifiability results. Hälvä et al. (2021); Klindt et al. (2020); Yao et al. (2022b;a); Lachapelle et al. (2022) further extend this nonlinear ICA framework into the scenarios of the time-delayed dynamical system, which allows the temporal transitions among the latent variables.

However, these nonlinear ICA-based methods usually assume that the mixing function (the generation process from sources to observations) is invertible, which may be difficult to satisfy in real-world scenarios, such as the 3D to 2D projection in the visual process. As shown in Figure 1 (a) and (b), we provide two intuitive instances of the real videos to illustrate how the non-invertibility happens. In (a), when object occlusions occur, information from the obstructed object is lost in the generation process of the current time step, which causes non-invertibility. In (b), the persistence of vision introduces non-invertibility, since the the mixing process of the current time step utilizes the history information. We further found that the violation of this inevitability assumption may cause the nonlinear ICA

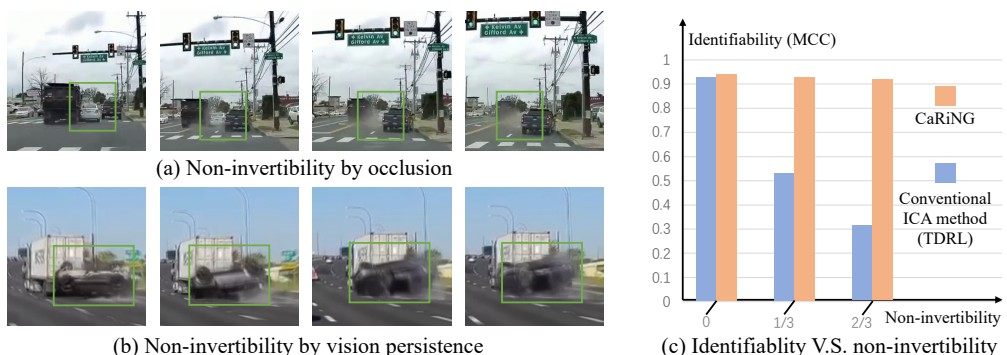

(a) Non-invertibility by occlusion

(b) Non-invertibility by vision persistence

(c) Identifiablity V.S. non-invertibility

Figure 1: **Motivations of the non-invertible generation process.** (a) The occlusions raise the non-invertibility since the measured observation cannot cover the obstructed objects. (b) The vision persistence, shown with the high-speed movement of a crashing car, describes the generation process that jointly involves the current state and previous, and causes the non-invertibility. (c) The identifiability of conventional methods, such as TDRL Yao et al. (2022a) (blue), drops drastically with the increase of non-invertibility. While our method's identifiability (orange) still holds. Here non-invertibility $= 1/3$ denotes the information loss is 1/3, i.e., the observed information (the number of observed variables) is 2/3 of the latent one (the number of latent variables).

method to obtain bad identification performance. In part (c) of Figure 1, we demonstrate that TDRL, one of the typical nonlinear ICA-based methods making the invertibility assumption, markedly degrades its performance in identifying the latent variables with increasing non-invertibility. It motivates us to extend the current nonlinear ICA to consider the non-invertible mixing function.

In this paper, to tackle the challenges above, we propose to leverage the temporal context aid in retrieving missing information caused by the non-invertible mixing function, mirroring the intuitive mechanisms of human perception. For instance, when we encounter an object with occlusion, our natural inclination is to draw from historical data to reconstruct the obscured portion. We demonstrate that, even when the generation process is non-invertible, the derived latent causal representation remains identifiable if the latent variables can be expressed as an arbitrary function combining the current observation with its history. Built upon this identification theorem, we introduce a principled approach, named **CaRiNG**, that learns the function to integrate historical data to compensate for the latent information lost due to non-invertibility. This approach extends the Sequential Variational Autoencoder (Sequential VAE) (Chung et al., 2015) with two distinct modifications. Firstly, it incorporates history (or context) information directly into the encoder. Specifically, we transform step-to-step mapping (from current observation to the current latent variable) into sequence-to-step mapping (from current observation and temporal context to the current latent variable). Secondly, a specialized prior module is introduced to determine the prior distribution of latent variables using the normalizing flow (Dinh et al., 2016), ensuring the imposition of an independent noise condition. We evaluate our method using both synthetic and real-world data. Using synthetic data, we design datasets with a non-invertible mixing function to gauge identifiability. For real-world applications, **CaRiNG** is deployed in a traffic accident reasoning task, a scenario wherein the intricate traffic dynamics introduce considerable non-invertibility. Experimental outcomes reveal that our method significantly outperforms other temporal representation learning methods for identifying causal representations amid non-invertible generation processes. Furthermore, this causal representation has proven instrumental in enhancing video reasoning tasks.

**Key Insights and Contributions** of our research include:

- To the best of our understanding, this paper presents the first identifiability theorem that accommodates a non-invertible generation process, which complements the existing body of the nonlinear ICA theory.

- We present a principled approach, **CaRiNG**, to learn the latent **Ca**usal **R**epresentation from the temporal data with **N**on-invertible **G**eneration processes with identifiability guarantees, by integrating the temporal context information to recover the lost information.

- Our evaluations across synthetic and real-world datasets attest to **CaRiNG**'s effectiveness for learning the identifiable latent causal representation, leading to enhancements in video reasoning tasks.

## 2 PROBLEM SETUP

### 2.1 NON-INVERTIBLE TEMPORAL GENERATIVE PROCESS

Consider observing $n$-dimensional time-series data with $T$ discrete time steps, represented as $\mathbf{X} = \{\mathbf{x}_1, \mathbf{x}_2, \ldots, \mathbf{x}_T\}$. Each observation $\mathbf{x}_t \in \mathcal{X} \subseteq \mathbb{R}^d$ is generated from a nonlinear mixing function $\mathbf{g}$ that maps $r + 1$ adjacent latent variables $\mathbf{z}_{t:t-r}$ to $\mathbf{x}_t$, where $\mathbf{z}_{t:t-r}$ refers to $\{\mathbf{z}_t, \mathbf{z}_{t-1}, \cdots, \mathbf{z}_{t-r}\}$. We have $\mathbf{z}_t \in \mathcal{Z} \subseteq \mathbb{R}^n$. For every $i \in 1, \ldots, n$, the variable $z_{it}$ of $\mathbf{z}_t$ is derived from a stationary, non-parametric time-delayed causal relation:

$$\underbrace{\mathbf{x}_t = \mathbf{g}(\mathbf{z}_{t:t-r})}_{\text{Nonlinear mixing}}, \quad \underbrace{z_{it} = f_i\left(\{z_{j,t-\tau} | z_{j,t-\tau} \in \mathbf{Pa}(z_{it})\}, \epsilon_{it}\right)}_{\text{Stationary non-parametric transition}} \; with \; \underbrace{\epsilon_{it} \sim p_{\epsilon_i}}_{\text{Stationary noise}} \; . \tag{1}$$

Note that with non-parametric causal transitions, the noise term $\epsilon_{it} \sim p_{\epsilon_i}$ (where $p_{\epsilon_i}$ denotes the distribution of $\epsilon_{it}$) and the time-delayed parents $\mathbf{Pa}(z_{it})$ of $z_{it}$ (i.e., the set of latent factors that directly cause $z_{it}$) are interacted and transformed in an arbitrarily nonlinear way to generate $z_{it}$. $\tau$ denotes the transition time lag. The components of $\mathbf{z}_t$ are mutually independent conditional on history variables $\mathbf{Pa}(\mathbf{z}_t)$.

In this case, one cannot recover $\mathbf{z}_t$ from $\mathbf{x}_t$ alone due to the non-invertibility of $\mathbf{g}$. Without extra assumptions, it is definitely non-identifiable. As a result, we assume that there exists a time lag $\mu$ and a nonlinear function $\mathbf{m}$ which can map a series of observations to latent variable $\mathbf{z}_t$, i.e.,

$$\mathbf{z}_t = \mathbf{m}(\mathbf{x}_{t:t-\mu}). \tag{2}$$

Once we successfully recover the information lost due to non-invertibility from the context, the classical nonlinear ICA algorithm can be used to solve this problem.

### 2.2 IDENTIFICATION OF THE LATENT CAUSAL PROCESSES

We define the identifiability of the latent causal process in **Definition 1**.

**Definition 1** (Identifiable Latent Causal Process). *Let $\mathbf{X} = \{\mathbf{x}_1, \mathbf{x}_2, \ldots, \mathbf{x}_T\}$ be a sequence of observed variables generated by the true temporally causal latent processes specified by $(f_i, p(\epsilon_i), \mathbf{g})$ given in Eq 1. A learned generative model $(\hat{f}_i, \hat{p}(\epsilon_i), \hat{\mathbf{g}})$ is observational equivalent to $(f_i, p(\epsilon_i), \mathbf{g})$ if the model distribution $p_{\hat{f}_i, \hat{p}_\epsilon, \hat{\mathbf{g}}}(\{\mathbf{x}_1, \mathbf{x}_2, \ldots, \mathbf{x}_T\})$ matches the data distribution $p_{f_i, p_\epsilon, \mathbf{g}}(\{\mathbf{x}_1, \mathbf{x}_2, \ldots, \mathbf{x}_T\})$ for any value of $\mathbf{x}_t$. We say latent causal processes are identifiable if observational equivalence can lead to a version of latent variable $\mathbf{z}_t = \mathbf{m}(\mathbf{x}_{t:t-\mu})$ up to permutation $\pi$ and component-wise invertible transformation $T$:*

$$p_{\hat{f}_i, \hat{p}_{\epsilon_i}, \hat{\mathbf{g}}}(\{\mathbf{x}_1, \mathbf{x}_2, \ldots, \mathbf{x}_T\}) = p_{f_i, p_{\epsilon_i}, \mathbf{g}}(\{\mathbf{x}_1, \mathbf{x}_2, \ldots, \mathbf{x}_T\})$$
$$\Rightarrow \hat{\mathbf{m}}(\mathbf{x}_{t:t-\mu}) = (T \circ \pi \circ \mathbf{m})(\mathbf{x}_{t:t-\mu}), \quad \forall \mathbf{x}_{t:t-\mu} \in \mathcal{X}^{\mu+1}, \tag{3}$$

*where $\mathcal{X}^{\mu+1}$ is the observation space.*

Different from the existing literature, we involve $\mathbf{m}$ in the above definition, since it does so implicitly as a property of the mixing function $\mathbf{g}$, although it does not explicitly participate in the generation process. Furthermore, the identifiability of $\mathbf{g}$ is different. In previous nonlinear ICA methods (Yao et al., 2022a; Hyvarinen & Morioka, 2017), the mixing function $\mathbf{g}$ is identifiable. However, in our case, we cannot find the identifiable mixing function since the information loss is caused by non-invertibility. Instead, we can obtain a component-wise transformation of a permuted version of latent variables $\hat{\mathbf{z}}_t = \mathbf{m}(\mathbf{x}_{t:t-\mu})$. The latent causal relations are also identifiable, up to a permutation $\pi$ and component-wise invertible transformation $T$, i.e., $\hat{\mathbf{f}} = T \circ \pi \circ \mathbf{f}$, once $\mathbf{z}_t$ is identifiable. It is because, in the time-delayed causally sufficient system, the conditional independence relations fully characterize time-delayed causal relations when we assume no latent causal confounders in the (latent) causal processes.

### 2.3 ILLUSTRATIONS OF THE PROBLEM SETUP

**Intuitive Illustration with Visual Persistence.** Consider a rapidly moving ball on a two-dimensional plane as described in figure 2. The horizontal and vertical coordinates of the ball's position at any given moment can be represented by the latent variable $\mathbf{z}_t \in \mathbb{R}^2$. We assume that the ball follows a curved trajectory constrained by the nonlinear function $\mathbf{f}$ as it moves.

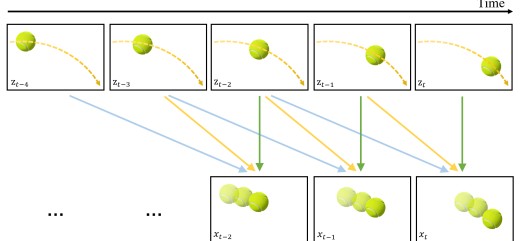

Suppose that we observe the ball with a visual persistence effect, where each observation $\mathbf{x}_t$ captures several consecutive latent variables as $\mathbf{x}_t = \mathbf{g}(\mathbf{z}_{<t})$. The mixing function $\mathbf{g}$ refers to the weighted sum of the images obtained through multiple exposures, which is what a person ultimately observes as $\mathbf{x}_t$. In this case, the invertibility of the mapping from $\mathbf{z}_t$ to $\mathbf{x}_t$ is compromised since the current frame also contains the latent information from previous frames.

Figure 2: **An intuitive illustration of a moving ball with a visual persistence effect.** The ball is moving along a descending curve. The observation in a particular time step is composed of multiple residual images of the ball.

**Mathematical Illustration.** Besides, we provide a mathematical example to demonstrate the existence of function $\mathbf{m}$ in Eq 2. Following the concept of visual persistence, let the current observation be a weakened previous observation overlaid with the current image of the object, i.e., $\mathbf{x}_t = \mathbf{z}_t + \frac{1}{2}\mathbf{x}_{t-1} = \sum_{i=1}^{\infty} \left(\frac{1}{2}\right)^i \mathbf{z}_{t-i}$ (Wolford, 1993). Given an extra observation, the current latent variable can be rewritten as $\mathbf{z}_t = \mathbf{x}_t - \frac{1}{2}\mathbf{x}_{t-1}$. Thereby we can easily recover latent variables that cannot be obtained from a single observation, i.e., $\mathbf{z}_t = \mathbf{m}(\mathbf{x}_{t:t-1}) = \mathbf{x}_t - \frac{1}{2}\mathbf{x}_{t-1}$.

## 3 IDENTIFIABILITY THEORY

In this section, we demonstrate that, given certain mild conditions, the learned causal representation $\mathbf{z}_t$ is identifiable up to permutation and a component-wise transformation. This holds even if the mixing function $\mathbf{g}$ is non-invertible. Firstly, we present the identifiability results when faced with a non-invertible mixing function and stationary transitions. Subsequently, we address the gap between permutation-scaling Jacobian to identifiability. Lastly, by leveraging side information such as the domain index and label, we illustrate how identifiability can be achieved even in a non-stationary context. The exhaustive proofs are available in Appendix A1.

### 3.1 IDENTIFIABILITY UNDER NON-INVERTIBLE GENERATIVE PROCESS

W.L.O.G., we first consider a simplified case with $\tau = r + 1$ and context length $\mu$, which infers such process:

$$\mathbf{x}_t = \mathbf{g}(\mathbf{z}_{t:t-r}), \quad z_{it} = f_i\left(\mathbf{z}_{t-1:t-r-1}, \epsilon_{it}\right), \tag{4}$$

where a function $\mathbf{m}$ satisfying $\mathbf{z}_t = \mathbf{m}(\mathbf{x}_{t:t-\mu})$ exists. When taking $r = 0$, the time delay is present only in transitions and is absent in the generation process. Taking $r > 0$ yields us to a more intricate scenario, where the mixing function encompasses not just the latent causal variables of the current time step, but also the information of previous steps, termed the Time-delayed Mixing Process. Such a scenario is compelling, acknowledging that the mixing process can be influenced by time-delayed effects. To illustrate, human visual perception provides a fitting example: the phenomenon known as the persistence of vision reveals that humans retain impressions of a visual stimulus even after its cessation (Coltheart, 1980). The extensions for any time lag $\tau$ will be discussed in Appendix A1.2.

**Theorem 1** (Identifiability under Non-invertible Generative Process). *For a series of observations* $\mathbf{x}_t$ *and estimated latent variables* $\hat{\mathbf{z}}_t$, *suppose there exists function* $\hat{\mathbf{g}}, \hat{\mathbf{m}}$ *which subject to observational equivalence, i.e.,*

$$\mathbf{x}_t = \hat{\mathbf{g}}(\hat{\mathbf{z}}_{t:t-r}), \quad \hat{\mathbf{z}}_t = \hat{\mathbf{m}}(\mathbf{x}_{t:t-\mu}). \tag{5}$$

*If assumptions*

- *(conditional independence) the components of* $\hat{\mathbf{z}}_t$ *are mutually independent conditional on* $\hat{\mathbf{z}}_{t-1:t-r-1}$,

- *(sufficiency) let $\eta_{kt} \triangleq \log p(z_{kt}|\mathbf{z}_{t-1:t-r-1})$, and*

$$\mathbf{v}_{k,t} \triangleq \left( \frac{\partial^2 \eta_{kt}}{\partial z_{k,t} \partial z_{1,t-r-1}}, \frac{\partial^2 \eta_{kt}}{\partial z_{k,t} \partial z_{2,t-r-1}}, ..., \frac{\partial^2 \eta_{kt}}{\partial z_{k,t} \partial z_{n,t-r-1}}, \mathbf{0}, \mathbf{0}, \cdots, \mathbf{0} \right)^{\mathsf{T}}$$

$$\mathring{\mathbf{v}}_{k,t} \triangleq \left( \mathbf{0}, \mathbf{0}, \cdots, \mathbf{0}, \frac{\partial^3 \eta_{kt}}{\partial z_{k,t}^2 \partial z_{1,t-r-1}}, \frac{\partial^3 \eta_{kt}}{\partial z_{k,t}^2 \partial z_{2,t-r-1}}, ..., \frac{\partial^3 \eta_{kt}}{\partial z_{k,t}^2 \partial z_{n,t-r-1}} \right)^{\mathsf{T}}. \tag{6}$$

  *for each value of $\mathbf{z}_t$, $\mathbf{v}_{1t}, \mathring{\mathbf{v}}_{1t}, \mathbf{v}_{2t}, \mathring{\mathbf{v}}_{2t}, ..., \mathbf{v}_{nt}, \mathring{\mathbf{v}}_{nt} \in \mathbf{R}^{2n}$, as $2n$ vector functions in $z_{1,t-1}$, $z_{2,t-1}, ..., z_{n,t-1}$, are linearly independent,*

- *(continuity) the domain of $\hat{\mathbf{z}}$ is path-connected, $\mathbf{m}, \hat{\mathbf{m}}, \mathbf{g}, \hat{\mathbf{g}}$ are second-order differentiable, and non-degeneracy condition holds for $\mathbf{m} \circ \hat{\mathbf{g}} \circ \hat{\mathbf{m}}$ and $\hat{\mathbf{m}} \circ \mathbf{g} \circ \mathbf{m}$,*

*are satisfied, then $\mathbf{z}_t$ must be a component-wise transformation of a permuted version of $\hat{\mathbf{z}}_t$ with regard to context $\{\mathbf{x}_j \mid \forall j = t, t-1, \cdots, t - \mu - r\}$.*

The proof of Theorem 1 can be found in Appendix A1.1. It is inspired from Yao et al. (2022a), which follows the line of Hyvarinen et al. (2019).

For clarification, the condition that a function $\mathbf{h} : \mathbb{R}^n \to \mathbb{R}^n$ is invertible, or equivalently the non-vanishing of the determinant of the Jacobian matrix $\mathbf{H}_h$, is called the non-degeneracy condition. We first define the partially invertible function, and then give the non-degeneracy condition on it.

**Definition 2** (Partially Invertiblility). *A function $\mathbf{z} = \mathbf{h}(\hat{\mathbf{z}}, \mathbf{c})$, where $\mathbf{z}, \hat{\mathbf{z}} \in \mathcal{Z} \subseteq \mathbb{R}^n$ and $\mathbf{z} \in \mathcal{C} \subseteq \mathbb{R}^m$, is partially invertible, if and only if for any given $\mathbf{c}$, the unfixed part $\mathbf{h}_{\mathbf{c}} : \mathbb{R}^n \to \mathbb{R}^n$ is always invertible.*

**Definition 3** (Non-degeneracy Condition of Partially Invertible Functions). *The non-degeneracy condition of a partially invertible function $\mathbf{z} = \mathbf{h}(\hat{\mathbf{z}}, \mathbf{c})$ is that for any given $\mathbf{c}$, the determinant of the Jacobian matrix $\mathbf{H}_{\mathbf{h}_{\mathbf{c}}}$ of $\mathbf{h}_{\mathbf{c}}$ is always non-zero.*

Besides, the nonstationary transition can also help to improve the identifiability of `CaRiNG` . As shown in the sufficiency assumption in Theorem 1, the identifiability relies on the sufficient changes of the conditional distribution $p(z_{kt}|\mathbf{z}_{t-1:t-r-1})$. When the distribution of the noise term varies between different domains, the domain index can serve as an auxiliary variable to improve this sufficiency since both domain dynamics and history variables can provide changes. A further discussion is provided in the Appendix A1.4.

## 3.2 NECESSITY OF CONTINUITY

To establish identifiability, numerous existing nonlinear ICA-based methods (Yao et al., 2022b;a; Hyvarinen et al., 2019; Hälvä et al., 2021) utilize the Jacobian matrix, denoted by $\mathbf{H} = \frac{\partial \mathbf{z}}{\partial \hat{\mathbf{z}}}$, which captures the relationship between ground truth and estimated latent variables. These methods propose that the learned latent variables are identifiable if $\mathbf{H}_{ij} \cdot \mathbf{H}_{ik} = 0$ for $j \neq k$ (with only a single non-zero element in each row or column). $\mathbf{H}$ corresponds to the Jacobian matrix of the function $\mathbf{h} \triangleq \mathbf{m} \circ \hat{\mathbf{g}}$ in our scenario (or $\mathbf{g}^{-1} \circ \hat{\mathbf{g}}$ for the general scenario). However, it is crucial to highlight an often overlooked shortcoming: this condition alone is insufficient to establish identifiability when dealing with non-linear generation processes. While in linear ICA, given that the Jacobian remains constant, this condition indeed equates to identifiability. Yet, in nonlinear ICA, the Jacobian matrix, being a function of $\hat{\mathbf{z}}$, can vary with different $\hat{\mathbf{z}}$ values, potentially rendering the mapping unpredictable. A comprehensive discussion is available in Appendix A1.3. We are happy to find that, concurrently to our work, Lachapelle et al. (2023) have rightfully pointed out that one has to be careful when going from "Jacobian is a permutation-scaling matrix" to "the mapping is a permutation composed with an element-wise transformation" when the domain of the function is not simply $\mathbf{R}^n$. Please refer to Lachapelle et al. (2023)'s discussion about "local" and "global" disentanglement.

To fill this gap, we provide two more assumptions. The domain $\hat{\mathcal{Z}}$ of $\hat{\mathbf{z}}$ should be path-connected, i.e., for any $\mathbf{a}, \mathbf{b} \in \hat{\mathcal{Z}}$, there exists a continuous path connecting $\mathbf{a}$ and $\mathbf{b}$ with all points of the path in $\hat{\mathcal{Z}}$. In addition, function $\mathbf{h}$ is second-order differentiable and holds the non-degeneracy condition. Without this condition, the map may or may not be locally invertible.

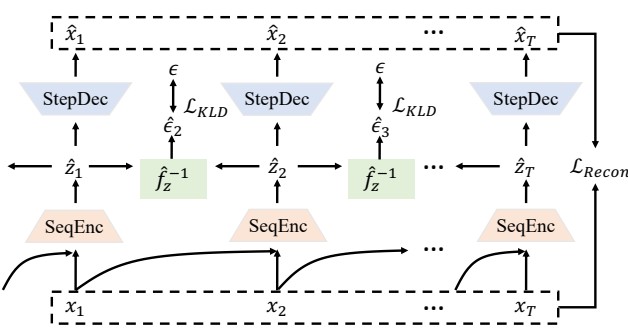

Figure 3: **The overall framework of `CaRiNG`.** It consists of three main modules, including the sequence-to-step encoder, step-to-step decoder, and the transition prior module, which is represented as SeqEnc, StepDec, and $\hat{\mathbf{f}}_{\mathbf{z}}^{-1}$ in a different color, respectively. The model is trained with both $\mathcal{L}_{Recon}$ and $\mathcal{L}_{KLD}$.

**Lemma 1** (Disentanglement with Continuity). *For second-order differentiable invertible function $\mathbf{h}$ defined on a path-connected domain $\hat{\mathcal{Z}} \subseteq \mathbb{R}^n$ which satisfies $\mathbf{z} = \mathbf{h}(\hat{\mathbf{z}})$, suppose the non-degeneracy condition holds. If there exists at most one non-zero entry in each row of the Jacobian matrix $\mathbf{H} = \frac{\partial \mathbf{z}}{\partial \hat{\mathbf{z}}}$, $\hat{\mathbf{z}}$ is a disentangled version of $\mathbf{z}$ up to a permutation and a element-wise nonlinear operation.*

Furthermore, when the Jacobian matrix is more than a function of $\hat{\mathbf{z}}$, but also is influenced by a side information $\mathbf{c}$, the identifiability can be guaranteed as well under mild extra conditions. We leverage Lemma 2 for the proof of Theorem 1.

**Lemma 2** (Disentanglement with Continuity under Side Information). *For second-order differentiable invertible function $\mathbf{h}$ defined on a path-connected domain $\hat{\mathcal{Z}} \times \mathcal{C} \subseteq \mathbb{R}^{n+m}$ which satisfies $\mathbf{z} = \mathbf{h}(\hat{\mathbf{z}}, \mathbf{c})$, suppose the non-degeneracy condition holds. If there exists at most one non-zero entry in each row of the Jacobian matrix $\mathbf{H}(\mathbf{c}) = \frac{\partial \mathbf{z}}{\partial \hat{\mathbf{z}}}$, $\hat{\mathbf{z}}$ is a disentangled version of $\mathbf{z}$ up to a permutation and a element-wise nonlinear operation.*

## 4 APPROACH

Given our results on identifiability, we introduce our `CaRiNG` approach. This aims to estimate the latent causal dynamics presented in Eq 1, even when faced with a non-invertible mixing procedure. To achieve this, `CaRiNG` builds upon the Sequential Variational Auto-Encoders (Li & Mandt, 2018) and incorporates three primary modules: the sequence-to-step encoder (SeqEnc), the step-to-step decoder (StepDec), and the transition prior module ($\hat{\mathbf{f}}_{\mathbf{z}}^{-1}$). During the training phase, we integrate the conditions from Sec. 3 as constraints and adopt two corresponding loss functions.

**Overall Framework.** As visualized in Figure 3, our framework starts by acquiring the latent causal representation via a sequence-to-step encoder, whose input and output are a sequence of observations $\mathbf{x}_{t:t-\mu}$ and the estimated latent variable $\hat{\mathbf{z}}_t$. Formally, it denotes the inference process of $q(\hat{\mathbf{z}}_t | \mathbf{x}_{t:t-\mu})$, which is corresponding to the function $\mathbf{m}$ in Eq 2. Following this, observations are generated from the latent space through a step-to-step decoder $p(\hat{\mathbf{x}}_t | \hat{\mathbf{z}}_t)$, which implies the mixing function $\mathbf{g}$ as mentioned in Eq 1. To learn the independent latent variables, we apply a constraint using the KL divergence between the posterior distribution of learned latent variables and a prior distribution which subjects to our conditional independence assumption in Theorem 1. The estimation of the prior distribution motivates us to utilize a normalizing flow, converting the prior distribution into Gaussian noise, represented as $\hat{\epsilon}_{it} = f_i^{-1}(\hat{z}_{it}, \hat{\mathbf{z}}_{t-1:t-\tau})$. Moreover, a reconstruction loss between the ground truth and generated observations is integrated for model training. A detailed exploration of all modules and losses is forthcoming.

**Sequence-to-Step Encoder and Step-to-Step Decoder.** Drawing inspiration from the capability of the human visual system, we utilize temporal context to reclaim the information lost due to non-invertible generation. The human visual system adeptly fills in occluded segments by recognizing coherent motion cues (Palmer, 1999; Wertheimer, 1938; Spelke, 1990). Assuming the presence of a function that captures all latent information from the current observation and its temporal context, we can retrieve the latent causal process with identifiability, i.e. $\mathbf{m}$ exists. Various non-linear models are suitable for estimating this function, taking a sequence of observations, $\mathbf{x}_{t:t-\mu}$, with a lag of $\mu$ as inputs, and yielding the estimated latent representation for the current time step as output. In our experiments, we utilize both Multi-Layer Perceptron (MLP) (Werbos, 1974) and Transformer (Vaswani et al., 2017) models, catering to different complexities. Given the estimated

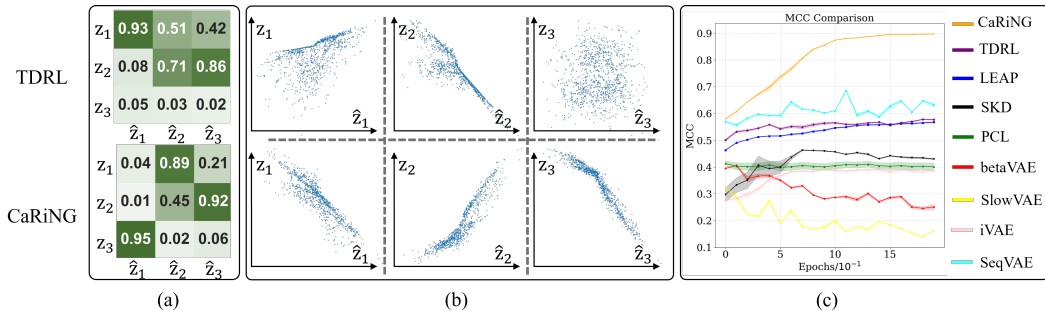

(a)                          (b)                          (c)

Figure 4: **Qualitative comparisons between baseline methods (especially TDRL) and CaRiNG in the setting of Non-invertible Generation.** (a) MCC matrix for all 3 latent variables; (b) The scatter plots between the estimated and ground-truth latent variables (only the aligned variables are plot); (c) The validation MCC curves of CaRiNG and other baselines.

latent variable $\hat{\mathbf{z}}_t$, a step-to-step decoder is employed to generate the current observation $\mathbf{x}_t$. For practical implementation, an MLP network suffices.

**Transition Prior Module.** To uphold the conditional independence assumption, we propose to minimize the KL divergence between the posterior distribution, $q(\hat{\mathbf{z}}_t|\mathbf{x}_{t:t-\mu})$, and a prior distribution $p(\hat{\mathbf{z}}_t|\hat{\mathbf{z}}_{t-1:t-\tau})$. By hard-coding the prior distribution to be conditionally independent on $\hat{\mathbf{z}}_{t-1:t-\tau}$, we expect the posterior to also be subject to the conditional independence assumption. Direct estimation of the prior, which has an arbitrary density function, poses challenges. As a solution, we introduce a transition prior module that facilitates the estimation of the prior using normalizing flow. Specifically, the prior is represented through a Gaussian distribution combined with the Jacobian matrix of the transition module.

Formally presented, the transition prior module is represented as $\hat{\epsilon}_{it} = \hat{f}_i^{-1}(\hat{z}_{it}, \hat{\mathbf{z}}_{t-1:t-\tau})$. Subsequently, the joint distribution is decomposed as a product of the noise distribution and the determinant of the Jacobian matrix, formulated as $p([\hat{\mathbf{z}}_{t-1:t-\tau}, \hat{\mathbf{z}}_t]) = p([\hat{\mathbf{z}}_{t-1:t-\tau}, \hat{\epsilon}_t]) \times |\mathbf{J}|$, with $\mathbf{J} = \begin{bmatrix} \mathbb{I}_{n\tau} & \mathbf{0} \\ \mathbf{0} & diag(\frac{\partial \hat{\epsilon}_{it}}{\partial \hat{z}_{it}}) \end{bmatrix}$, where $[\cdot]$ denotes concatenation operation. Leverage this joint distribution, we can derive the prior as

$$
\begin{aligned}
\log p(\hat{\mathbf{z}}_t|\hat{\mathbf{z}}_{t-1:t-\tau}) &= \log p([\hat{\mathbf{z}}_t, \hat{\mathbf{z}}_{t-1:t-\tau}]) - \log p(\hat{\mathbf{z}}_{t-1:t-\tau}) \\
&= \log p([\hat{\epsilon}_t, \hat{\mathbf{z}}_{t-1:t-\tau}]) + \log |\mathbf{J}| - \log p(\hat{\mathbf{z}}_{t-1:t-\tau}) \\
&= \log p(\hat{\epsilon}_t|\hat{\mathbf{z}}_{t-1:t-\tau}) + \log |\mathbf{J}| \\
&= \log p(\hat{\epsilon}_t) + \log |\mathbf{J}| \\
&= \sum_i \log p(\hat{\epsilon}_{it}) + \log |\mathbf{J}| \qquad \text{: Conditional independence assumption} \\
&= \sum_i \left( \log p(\hat{\epsilon}_{it}) + \log \frac{\partial \hat{\epsilon}_{it}}{\partial \hat{z}_{t,i}} \right) \qquad \text{: Lower-triangular Jacobian.}
\end{aligned}
$$

(7)

The transition prior module can be efficiently executed using an MLP network, transforming the latent variables $\hat{\mathbf{z}}_{t:t-\tau}$ into $\hat{\epsilon}_t$.

**Optimization.** We train CaRiNG using the Evidence Lower BOund (ELBO) objective, which is written as follows:

$$
\begin{aligned}
\text{ELBO} &\triangleq \mathbb{E}_{q_\phi(\mathbf{Z}|\mathbf{X})}[\log p_\theta(\mathbf{X}|\mathbf{Z})] - D_{KL}(q_\phi(\mathbf{Z}|\mathbf{X})||p_\theta(\mathbf{Z})) \\
&= \underbrace{\mathbb{E}_{q_\phi(\mathbf{Z}|\mathbf{X})} \sum_{t=1}^T \log p_\theta(\mathbf{x}_t|\mathbf{z}_t)}_{-\mathcal{L}_{\text{Recon}}} + \underbrace{\mathbb{E}_{q_\phi(\mathbf{Z}|\mathbf{X})} \left[ \sum_{t=1}^T \log p_\theta(\mathbf{z}_t|\mathbf{z}_{t-1:t-\tau}) - \sum_{t=1}^T \log q_\phi(\mathbf{z}_t|\mathbf{x}_{t:t-\mu}) \right]}_{-\mathcal{L}_{\text{KLD}}}.
\end{aligned}
$$

(8)

For the reconstruction likelihood $\mathcal{L}_{\text{Recon}}$, we utilize the mean-squared error (MSE) to measure the discrepancy between the generated and original observations. When computing the KL divergence $\mathcal{L}_{\text{KLD}}$, we resort to a sampling method, given that the prior distribution lacks an explicit form. To elaborate, the posterior is produced by the encoder, while the prior is defined as in Eq 7.

Table 1: **MCC scores (with standard deviations over 4 random seeds) of `CaRiNG` and other baselines on both NG and NG-TDMP settings .**

| Settings | Method | | | | | | | | |
|---|---|---|---|---|---|---|---|---|---|
| | **CaRiNG** | TDRL | LEAP | SlowVAE | PCL | betaVAE | SKD | iVAE | SequentialVAE |
| NG | **0.933 ±0.010** | 0.627 ±0.009 | 0.651 ±0.019 | 0.362 ±0.041 | 0.507 ±0.091 | 0.551 ±0.007 | 0.489 ±0.077 | 0.391 ±0.686 | 0.750 ±0.035 |
| NG-TDMP | **0.921 ±0.010** | 0.837 ±0.068 | 0.704 ±0.005 | 0.398 ±0.037 | 0.489 ±0.095 | 0.437 ±0.021 | 0.381 ±0.084 | 0.553 ±0.097 | 0.847 ±0.019 |

## 5 EXPERIMENTS

We conducted the experiments in two simulated environments, utilizing the available ground truth latent variables to evaluate identifiability. Subsequently, we assessed **CaRiNG** on a real-world VideoQA task, SUTD-TrafficQA (Xu et al., 2021), to verify its capability in representing complex and non-invertible traffic events.

### 5.1 SIMULATION EXPERIMENTS

**Dataset and experimental settings.** To evaluate whether **CaRiNG** can learn the causal process and identify the latent variables under a non-invertible scenario, we design a series of simulation experiments based on a random causal structure with a given sample size and variable size. We provide two experimental settings, including NG and NG-TDMP, which simulate the scenarios in Theorem 1 with $r = 0$ (non-invertible generation) and $r > 0$ (time-delayed mixing process), respectively. In particular, for NG, we simulate the visual perception system that uses the ground-truth dimension as 3 to represent the 3D real world and apply 2 measured variables to represent the 2D observation, which indicates the generation is non-invertible. For NG-TDMP, we simulate the persistence of vision that involves the previous latent variables in the current mixing process. It denotes that even if the dimension of the observation is not reduced, the generation process is still non-invertible due to the time-delay mixing. More details of the data generation process can be found in the Appendix.

**Evaluation metrics.** We apply the standard evaluation metric in the field of ICA, Mean Correlation Coefficient (MCC), to evaluate the identifiability of our **CaRiNG**. MCC measures the recovery of latent factors by calculating the absolute values of the correlation coefficient between every ground-truth factor against every estimated latent variable. It first calculates the Pearson correlation coefficients to measure the relationship and then adjusts the order with an assignment algorithm. The MCC score is a value from 0 to 1, where higher is with better identifiability.

**Baseline methods.** We compare **CaRiNG** with a series of baseline methods. BetaVAE (Higgins et al., 2017) is the most basic baseline with no identifiability guarantee. SlowVAE (Klindt et al., 2020), and PCL (Hyvarinen & Morioka, 2017) assume the independent sources even utilize the temporal information. LEAP (Yao et al., 2022b) and TDRL (Yao et al., 2022a) allow for learning causal processes but assume an invertible generation process. Besides, we also compare **CaRiNG** with the disentangled representation learning methods, such as SKD (Berman et al., 2022), which are not based on ICA and don't have the identifiability guarantee. In addition, we compare with iVAE (Khemakhem et al., 2020), despite iVAE not theoretically working under stationary transition. It is important to note that iVAE requires additional domain information as input. In our experiments, we simply used time indices as the domain label. Lastly, Sequential VAE (Chung et al., 2015) is also compared to verify the effect of conditional independence.

**Quantitative results.** The performance of **CaRiNG** and other baseline methods in both the NG and NG-TDMP scenarios is presented in Table 1. Initially, it's evident that all baseline Nonlinear ICA methods yield unsatisfactory MCC scores in both scenarios, including the strong TDRL baseline, which previously obtained good results in invertible settings, as shown in Figure 4 (c). As shown in Figure 4 (a), TDRL cannot recover the lost latent variables caused by non-invertible generation (MCC=0.03 for that variable). It is also illustrated by the scatter plots in Figure 4 (b), which show the independence between the estimated and ground truth variables on that dimension. Interestingly, we find that the Sequential VAE method works better than other methods that don't use the temporal context, which also demonstrates the necessity of temporal context to solve the invertibility issue. However, we still find that constraining the conditional independence benefits better performance, which shows the effect of the KL part. Furthermore, **CaRiNG** consistently delivers robust identifiability outcomes in both settings. This suggests that leveraging temporal context significantly enhances identifiability when faced with non-invertible generation processes. Lastly, performance in the NG scenario is better than that in the NG-TDMP scenario, showing the increased complexity introduced by the time-delayed mixing process.

Table 2: **Results on SUTD-TrafficQA dataset on different question types.** B: "Basic understanding", F: "Forecasting task", R: "Reverse Reasoning", C: "Counterfactual inference", I: "Introspection", A: "Attribution". The "ALL" column shows the weighted average of scores, based on each question type's population. The cross-modality matching parts of TDRL and `CaRiNG` are based on HCRN.

| Method | Year | Question Type | | | | | | All |
| --- | --- | --- | --- | --- | --- | --- | --- | --- |
| | | B | A | I | C | F | R | |
| I3D+LSTM | CVPR2017 | - | - | - | - | - | - | 33.21 |
| HCRN | CVPR2020 | 34.17 | 50.29 | 33.40 | 40.73 | 44.58 | 50.09 | 36.26 |
| VQAC | ICCV2021 | 34.02 | 49.43 | 34.44 | 39.74 | 38.55 | 49.73 | 36.00 |
| MASN | ACL2021 | 33.83 | 50.86 | 34.23 | 41.06 | 41.57 | 50.80 | 36.03 |
| DualVGR | TMM2021 | 33.91 | 50.57 | 33.40 | 41.39 | 41.57 | 50.62 | 36.07 |
| Eclipse | CVPR2021 | - | - | - | - | - | - | 37.05 |
| CMCIR | TPAMI2023 | 36.10 | **52.59** | **38.38** | 46.03 | **48.80** | **52.21** | 38.58 |
| TDRL | NeurIPS2022 | 36.28 | 39.57 | 29.63 | 46.49 | 31.84 | 39.46 | 37.32 |
| `CaRiNG` | - | **38.95** | 44.98 | 32.43 | **48.64** | 41.70 | 47.16 | **41.22** |

## 5.2 REAL-WORLD EXPERIMENTS

**Dataset and experimental settings.** The SUTD-TrafficQA dataset (Xu et al., 2021) is a comprehensive resource tailored for video event understanding in traffic scenarios, notably characterized by numerous occlusions among traffic agents. It consists of 10,090 videos and provides over 62,535 human-annotated QA pairs. Among them, 56,460 QA pairs are used for training and the rest 6,075 QA pairs are used for testing. The dataset challenges models with six key reasoning tasks: "Basic Understanding" is designed for grasping essential traffic dynamics. "Event Forecasting" and "Reverse Reasoning" evaluate the temporal prediction ability. "Introspection", "Attribution", and "Counterfactual Inference" require the model to understand the causal dynamic and conduct reasoning. All tasks are formulated as multiple-choice forms (evaluation with accuracy) without limiting the number of candidate answers, and demand a deep comprehension of traffic events and their underlying causality.

**Baseline methods.** The primary method we benchmark against is TDRL (Yao et al., 2022a), to evaluate the representation ability of the complex and non-invertible traffic environment. Additionally, we evaluate `CaRiNG` in comparison with state-of-the-art VideoQA methods, including I3D+LSTM (Carreira & Zisserman, 2017), HCRN (Le et al., 2020), VQAC (Kim et al., 2021), MASN (Seo et al., 2021), DualVGR (Wang et al., 2021), Eclipse (Xu et al., 2021), and CMCIR (Liu et al., 2023). In our approach, `CaRiNG` is leveraged to identify latent causal dynamics, while HCRN serves as the basic model for question answering. Further implementation details are provided in the Appendix.

**Quantitative results.** Performance comparisons for the six question types on SUTD-TrafficQA are summarized in Table 2. `CaRiNG` achieves a score of 41.22, which demonstrates a significant improvement which is nearly 6.8% over the next best method. Notably, when compared to TDRL, which lacks temporal context, `CaRiNG` exhibits significant advancements in representing complex, non-invertible traffic events. When benchmarked against the HCRN baseline, which employs the same cross-modality matching module, our approach further escalates the score by 4.96 through causal representation learning. Though CMCIR (Liu et al., 2023) applies the Swin-Transformer-L (Liu et al., 2021) pretrained on ImageNet-22K dataset as the frame-level appearance extractor and employs the video Swin-B (Liu et al., 2022) pretrained on Kinetics-600 as the clip-level motion feature extractor (more powerful than ours), `CaRiNG` with sample ResNet101 (He et al., 2016) features still outperforms it with 2.64 in average.

## 6 CONCLUSION

In this paper, we have proposed to consider learning temporal causal representation under the non-invertible generation process. We have established identifiability theories that allow for recovering the latent causal process with the nonlinear and non-invertible mixing function. Furthermore, based on this theorem, we proposed our approach, `CaRiNG`, to leverage the temporal context to estimate the lost latent information. We have conducted a series of simulated experiments to verify the identifiability results of `CaRiNG` under the non-invertible generations, and evaluated the learned representation in a complex and non-invertible traffic environment with real-world VideoQA tasks.

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

*Appendix for*

## "Learning Temporal Causal Representation under Non-Invertible Generation Process"

Appendix organization:

## A1 IDENTIFIABILITY THEORY

### A1.1 PROOF FOR THEOREM 1

Let us first shed light on the identifiability theory on the special case with $\tau = r + 1$, i.e.,

$$\mathbf{x}_t = \mathbf{g}(\mathbf{z}_{t:t-r}), \quad z_{it} = f_i\left(\mathbf{z}_{t-1}, \epsilon_{it}\right), \quad \mathbf{z}_t = \mathbf{m}(\mathbf{x}_{t:t-\mu}). \tag{9}$$

**Theorem A1** (Identifiability under Non-invertible Generative Process). *For a series of observations* $\mathbf{x}_t$ *and estimated latent variables* $\hat{\mathbf{z}}_t$, *suppose there exists function* $\hat{\mathbf{g}}, \hat{\mathbf{m}}$ *which subject to observational equivalence, i.e.,*

$$\mathbf{x}_t = \hat{\mathbf{g}}(\hat{\mathbf{z}}_{t:t-r}), \quad \hat{\mathbf{z}}_t = \hat{\mathbf{m}}(\mathbf{x}_{t:t-\mu}). \tag{10}$$

*If assumptions*

- *(conditional independence) the components of* $\hat{\mathbf{z}}_t$ *are mutually independent conditional on* $\hat{\mathbf{z}}_{t-1:t-r-1}$,

- *(sufficiency) let* $\eta_{kt} \triangleq \log p(z_{kt}|\mathbf{z}_{t-1:t-r-1})$, *and*

$$\mathbf{v}_{k,t} \triangleq \left(\frac{\partial^2 \eta_{kt}}{\partial z_{k,t}\partial z_{1,t-r-1}}, \frac{\partial^2 \eta_{kt}}{\partial z_{k,t}\partial z_{2,t-r-1}}, ..., \frac{\partial^2 \eta_{kt}}{\partial z_{k,t}\partial z_{n,t-r-1}}, \mathbf{0}, \mathbf{0}, \cdots, \mathbf{0}\right)^{\mathsf{T}}$$

$$\mathring{\mathbf{v}}_{k,t} \triangleq \left(\mathbf{0}, \mathbf{0}, \cdots, \mathbf{0}, \frac{\partial^3 \eta_{kt}}{\partial z_{k,t}^2 \partial z_{1,t-r-1}}, \frac{\partial^3 \eta_{kt}}{\partial z_{k,t}^2 \partial z_{2,t-r-1}}, ..., \frac{\partial^3 \eta_{kt}}{\partial z_{k,t}^2 \partial z_{n,t-r-1}}\right)^{\mathsf{T}}, \tag{11}$$

*for each value of* $\mathbf{z}_t$, $\mathbf{v}_{1t}, \mathring{\mathbf{v}}_{1t}, \mathbf{v}_{2t}, \mathring{\mathbf{v}}_{2t}, ..., \mathbf{v}_{nt}, \mathring{\mathbf{v}}_{nt} \in \mathbf{R}^{2n}$, *as* $2n$ *vector functions in* $z_{1,t-1}, z_{2,t-1}, ..., z_{n,t-1}$, *are linearly independent,*

- *(continuity) the domain of* $\hat{\mathbf{z}}$ *is path-connected, and* $\mathbf{m}, \hat{\mathbf{m}}, \mathbf{g}, \hat{\mathbf{g}}$ *are second-order differentiable, non-degeneracy condition holds for* $\mathbf{m} \circ \hat{\mathbf{g}} \circ \hat{\mathbf{m}}$ *and* $\hat{\mathbf{m}} \circ \mathbf{g} \circ \mathbf{m}$,

*are satisfied, then* $\mathbf{z}_t$ *must be a component-wise transformation of a permuted version of* $\hat{\mathbf{z}}_t$ *with regard to context* $\{\mathbf{x}_j \mid \forall j = t, t-1, \cdots, t-\mu-r\}$.

For a better understanding of the sufficiency assumption in Eq 11, we will now proceed to provide an explanation for it. Consider the mixed derivative matrix $\mathbb{V}_t, \mathring{\mathbb{V}}_t$ formed by $\{\mathbf{v}_{k,t}\}_{k=1}^n$, $\{\mathring{\mathbf{v}}_{k,t}\}_{k=1}^n$ as

$$\mathbb{V}_t = \begin{bmatrix} \frac{\partial^2 \eta_{1t}}{\partial z_{1,t}\partial z_{1,t-r-1}} & \frac{\partial^2 \eta_{1t}}{\partial z_{1,t}\partial z_{2,t-r-1}} & \cdots & \frac{\partial^2 \eta_{1t}}{\partial z_{1,t}\partial z_{n,t-r-1}} \\ \frac{\partial^2 \eta_{2t}}{\partial z_{2,t}\partial z_{1,t-r-1}} & \frac{\partial^2 \eta_{2t}}{\partial z_{2,t}\partial z_{2,t-r-1}} & \cdots & \frac{\partial^2 \eta_{2t}}{\partial z_{2,t}\partial z_{n,t-r-1}} \\ \vdots & \vdots & \ddots & \vdots \\ \frac{\partial^2 \eta_{nt}}{\partial z_{n,t}\partial z_{1,t-r-1}} & \frac{\partial^2 \eta_{nt}}{\partial z_{n,t}\partial z_{2,t-r-1}} & \cdots & \frac{\partial^2 \eta_{nt}}{\partial z_{n,t}\partial z_{n,t-r-1}} \end{bmatrix}^T \tag{12}$$

and

$$\mathring{\mathbb{V}}_t = \begin{bmatrix} \frac{\partial^3 \eta_{1t}}{\partial^2 z_{1,t}\partial z_{1,t-r-1}} & \frac{\partial^3 \eta_{1t}}{\partial^2 z_{1,t}\partial z_{2,t-r-1}} & \cdots & \frac{\partial^3 \eta_{1t}}{\partial^2 z_{1,t}\partial z_{n,t-r-1}} \\ \frac{\partial^3 \eta_{2t}}{\partial^2 z_{2,t}\partial z_{1,t-r-1}} & \frac{\partial^3 \eta_{2t}}{\partial^2 z_{2,t}\partial z_{2,t-r-1}} & \cdots & \frac{\partial^3 \eta_{2t}}{\partial^2 z_{2,t}\partial z_{n,t-r-1}} \\ \vdots & \vdots & \ddots & \vdots \\ \frac{\partial^3 \eta_{nt}}{\partial^2 z_{n,t}\partial z_{1,t-r-1}} & \frac{\partial^3 \eta_{nt}}{\partial^2 z_{n,t}\partial z_{2,t-r-1}} & \cdots & \frac{\partial^3 \eta_{nt}}{\partial^2 z_{n,t}\partial z_{n,t-r-1}} \end{bmatrix}^T \tag{13}$$

separately. The sufficiency assumption is satisfied if and only if $\mathbb{V}_t, \mathring{\mathbb{V}}_t$ are of full rank, thus $\begin{bmatrix} \mathbb{V}_t & \mathbf{0} \\ \mathbf{0} & \mathring{\mathbb{V}}_t \end{bmatrix}$ is of full rank. The purpose of this assumption is that the model can capture and distinguish independent noise from transition dynamics only when they are sufficiently diverse. This property will be used in the following proof.

*Proof.* For any $t$, combining Eq 9 and Eq 10 gives

$$
\begin{aligned}
\mathbf{z}_t &= \mathbf{m}(\mathbf{x}_{t:t-\mu}) \\
&= \mathbf{m}(\hat{\mathbf{g}}(\hat{\mathbf{z}}_t, \hat{\mathbf{z}}_{t-1:t-r}), \mathbf{x}_{t-1:t-\mu}) \\
&= \mathbf{m}(\hat{\mathbf{g}}(\hat{\mathbf{z}}_t, \hat{\mathbf{m}}(\mathbf{x}_{t-1:t-\mu-1}), \cdots, \hat{\mathbf{m}}(\mathbf{x}_{t-r:t-\mu-r})), \mathbf{x}_{t-1:t-\mu}),
\end{aligned}
\tag{14}
$$

as well as $\hat{\mathbf{z}}_t = \hat{\mathbf{m}}(\mathbf{g}(\mathbf{z}_t, \mathbf{m}(\mathbf{x}_{t-1:t-\mu-1}), \cdots, \mathbf{m}(\mathbf{x}_{t-r:t-\mu-r})), \mathbf{x}_{t-1:t-\mu})$ similarly. Upon Eq 14, we have an unified partially invertible function $\mathbf{z}_t = \mathbf{h}(\hat{\mathbf{z}}_t | \mathbf{x}_{t-1:t-\mu-r})$ where $\mathbf{h} = \mathbf{m} \circ \hat{\mathbf{g}}$ with Jacobian $\frac{\partial \mathbf{z}_t}{\partial \hat{\mathbf{z}}_t} = \mathbf{H}_t(\hat{\mathbf{z}}_t; \mathbf{x}_{t-1:t-\mu-r})$. By *partially invertible* it means that $\mathbf{z}$ and $\hat{\mathbf{z}}$ are in one-to-one correspondence for any context observations $\mathbf{x}_{t-1:t-\mu-r}$ that are fixed. One more thing to notify is that since $\mathbf{g}, \hat{\mathbf{g}}, \mathbf{m}, \hat{\mathbf{m}}$ are second-order differentiable, the nested $\mathbf{h}$ is also second-order differentiable. Let us consider the mapping from joint distribution $(\hat{\mathbf{z}}_t, \mathbf{x}_{t-1:t-\mu-r-1})$ to $(\mathbf{z}_t, \mathbf{x}_{t-1:t-\mu-r-1})$, i.e.,

$$
P(\mathbf{z}_t, \mathbf{x}_{t-1:t-\mu-r-1}) = P(\hat{\mathbf{z}}_t, \mathbf{x}_{t-1:t-\mu-r-1}) / |\mathbf{J}_t|,
\tag{15}
$$

where

$$
\mathbf{J}_t = \begin{bmatrix} \frac{\partial \mathbf{z}_t}{\partial \hat{\mathbf{z}}_t} & \mathbf{0} \\ * & \mathbf{I} \end{bmatrix},
\tag{16}
$$

which is a lower triangle matrix, where $\mathbf{I}$ infers eye matrix and $*$ infers any possible matrix. Thus, we have determinant $|\mathbf{J}_t| = |\frac{\partial \mathbf{z}_t}{\partial \hat{\mathbf{z}}_t}| = |\mathbf{H}_t|$. Dividing both sides of Eq 15 by $P(\mathbf{x}_{t-1:t-\mu-r-1})$ gives

$$
\mathbf{LHS} = P(\mathbf{z}_t | \mathbf{x}_{t-1:t-\mu-r-1}) = P(\mathbf{z}_t | \mathbf{z}_{t-1:t-r-1}),
\tag{17}
$$

since $\mathbf{z}_t$ and $\mathbf{x}_{t-1:t-\mu-r-1}$ are independent conditioned on $\mathbf{z}_{t-1:t-r-1}$. Similarly, $\mathbf{RHS} = P(\hat{\mathbf{z}}_t | \mathbf{x}_{t-1:t-\mu-r-1}) = P(\hat{\mathbf{z}}_t | \hat{\mathbf{z}}_{t-r-1})$ holds true as well, which yields to

$$
P(\mathbf{z}_t | \mathbf{z}_{t-1:t-r-1}) = P(\hat{\mathbf{z}}_t | \hat{\mathbf{z}}_{t-1:t-r-1}) / |\mathbf{H}_t|.
\tag{18}
$$

From a direct observation, if the components of $\hat{\mathbf{z}}_t$ are mutually independent given $\hat{\mathbf{z}}_{t-1:t-r-1}$, then for any distinct $i \neq j$, $\hat{z}_{it}$ and $\hat{z}_{jt}$ are conditionally independent given $(\hat{\mathbf{z}}_t \setminus \{\hat{z}_{it}, \hat{z}_{jt}\}) \cup \hat{\mathbf{z}}_{t-1:t-r-1}$. This mutual independence of the components of $\hat{\mathbf{z}}_t$ based on $\hat{\mathbf{z}}_{t-1:t-r-1}$ implies two things:

- $\hat{z}_{it}$ is independent from $\hat{\mathbf{z}}_t \setminus \{\hat{z}_{it}, \hat{z}_{jt}\}$ conditional on $\hat{\mathbf{z}}_{t-1:t-r-1}$. Formally,
$$
p(\hat{z}_{it} \,|\, \hat{\mathbf{z}}_{t-1:t-r-1}) = p(\hat{z}_{it} \,|\, (\hat{\mathbf{z}}_t \setminus \{\hat{z}_{it}, \hat{z}_{jt}\}) \cup \hat{\mathbf{z}}_{t-1:t-r-1}).
$$

- $\hat{z}_{it}$ is independent from $\hat{\mathbf{z}}_t \setminus \{\hat{z}_{it}\}$ conditional on $\hat{\mathbf{z}}_{t-1:t-r-1}$. Represented as:
$$
p(\hat{z}_{it} \,|\, \hat{\mathbf{z}}_{t-1:t-r-1}) = p(\hat{z}_{it} \,|\, (\hat{\mathbf{z}}_t \setminus \{\hat{z}_{it}\}) \cup \hat{\mathbf{z}}_{t-1:t-r-1}).
$$

From these two equations, we can derive:

$$
p(\hat{z}_{it} \,|\, (\hat{\mathbf{z}}_t \setminus \{\hat{z}_{it}\}) \cup \hat{\mathbf{z}}_{t-1:t-r-1}) = p(\hat{z}_{it} \,|\, (\hat{\mathbf{z}}_t \setminus \{\hat{z}_{it}, \hat{z}_{jt}\}) \cup \hat{\mathbf{z}}_{t-1:t-r-1}),
$$

which yields that $\hat{z}_{it}$ and $\hat{z}_{jt}$ are conditionally independent given $(\hat{\mathbf{z}}_t \setminus \{\hat{z}_{it}, \hat{z}_{jt}\}) \cup \hat{\mathbf{z}}_{t-1:t-r-1}$ for $i \neq j$. Leveraging an inherent fact, i.e., if $\hat{z}_{it}$ and $\hat{z}_{jt}$ are conditionally independent given $(\hat{\mathbf{z}}_t \setminus \{\hat{z}_{it}, \hat{z}_{jt}\}) \cup \hat{\mathbf{z}}_{t-1:t-r-1}$, the subsequent equation arises:

$$
\frac{\partial^2 \log p(\hat{\mathbf{z}}_t, \hat{\mathbf{z}}_{t-1:t-r-1})}{\partial \hat{z}_{it} \partial \hat{z}_{jt}} = 0,
$$

assuming the cross second-order derivative exists.

Given that $p(\hat{\mathbf{z}}_t, \hat{\mathbf{z}}_{t-1:t-r-1}) = p(\hat{\mathbf{z}}_t \,|\, \hat{\mathbf{z}}_{t-1:t-r-1}) p(\hat{\mathbf{z}}_{t-1:t-r-1})$ and $p(\hat{\mathbf{z}}_{t-1:t-r-1})$ remains independent of $\hat{z}_{it}$ or $\hat{z}_{jt}$, the above equality is equivalent to

$$
\frac{\partial^2 \log p(\hat{\mathbf{z}}_t \,|\, \hat{\mathbf{z}}_{t-1:t-r-1})}{\partial \hat{z}_{it} \partial \hat{z}_{jt}} = 0.
\tag{19}
$$

Referencing Eq 18, it gets expressed as:

$$
\log p(\hat{\mathbf{z}}_t \,|\, \hat{\mathbf{z}}_{t-1:t-r-1}) = \log p(\mathbf{z}_t \,|\, \mathbf{z}_{t-1:t-r-1}) + \log |\mathbf{H}_t| = \sum_{k=1}^{n} \eta_{kt} + \log |\mathbf{H}_t|.
\tag{20}
$$

The partial derivative w.r.t. $\hat{z}_{it}$ is presented below:

$$\frac{\partial \log p(\hat{\mathbf{z}}_t \mid \hat{\mathbf{z}}_{t-1:t-r-1})}{\partial \hat{z}_{it}} = \sum_{k=1}^{n} \frac{\partial \eta_{kt}}{\partial z_{kt}} \cdot \frac{\partial z_{kt}}{\partial \hat{z}_{it}} + \frac{\partial \log |\mathbf{H}_t|}{\partial \hat{z}_{it}}$$

$$= \sum_{k=1}^{n} \frac{\partial \eta_{kt}}{\partial z_{kt}} \cdot \mathbf{H}_{kit} + \frac{\partial \log |\mathbf{H}_t|}{\partial \hat{z}_{it}}.$$

The second-order cross derivative can be depicted as:

$$\frac{\partial^2 \log p(\hat{\mathbf{z}}_t \mid \hat{\mathbf{z}}_{t-1:t-r-1})}{\partial \hat{z}_{it} \partial \hat{z}_{jt}} = \sum_{k=1}^{n} \left( \frac{\partial^2 \eta_{kt}}{\partial z_{kt}^2} \cdot \mathbf{H}_{kit} \mathbf{H}_{kjt} + \frac{\partial \eta_{kt}}{\partial z_{kt}} \cdot \frac{\partial \mathbf{H}_{kit}}{\partial \hat{z}_{jt}} \right) + \frac{\partial^2 \log |\mathbf{H}_t|}{\partial \hat{z}_{it} \partial \hat{z}_{jt}}. \quad (21)$$

According to Eq 19, the right-hand side of the presented equation consistently equals 0. Therefore, for each index $l$ ranging from 1 to $n$, and every associated value of $z_{l,t-r-1}$, its partial derivative with respect to $z_{l,t-r-1}$ remains 0. That is,

$$\sum_{k=1}^{n} \left( \frac{\partial^3 \eta_{kt}}{\partial z_{kt}^2 \partial z_{l,t-r-1}} \cdot \mathbf{H}_{kit} \mathbf{H}_{kjt} + \frac{\partial^2 \eta_{kt}}{\partial z_{kt} \partial z_{l,t-r-1}} \cdot \frac{\partial \mathbf{H}_{kit}}{\partial \hat{z}_{jt}} \right) \equiv 0, \quad (22)$$

where we leveraged the fact that entries of $\mathbf{H}_t$ do not depend on $z_{l,t-r-1}$. Considering any given value of $\mathbf{z}_t, \mathbf{v}_{1t}, \mathring{\mathbf{v}}_{1t}, \mathbf{v}_{2t}, \mathring{\mathbf{v}}_{2t}, ..., \mathbf{v}_{nt}, \mathring{\mathbf{v}}_{nt}$ are linearly independent, to make the above equation hold true, one has to set $\mathbf{H}_{kit}\mathbf{H}_{kjt} = 0$ or $i \neq j$. In other words, each row of $\mathbf{H}_t$ consists of at most a single non-zero entry. Given that second-order differentiable function $\mathbf{h}$ is defined on path-connected domain and partially invertible with regard to context $\mathbf{x}_{t-1:t-\mu-r}$, in addition to non-degeneracy, $\mathbf{z}_t$ must be a component-wise transformation of a permuted version of $\hat{\mathbf{z}}_t$ with regard to context according to Lemma A2. □

Note that in the proof of Theorem A1, we require the transition lag $\tau$ to be larger than the mixing lag $r = 1$. When a mixing lag exists, the guarantee of identifiability requires dynamic information from a further previous time step. As long as this inequality $\tau > r$ is satisfied, the parameters $\tau$ can be extended to arbitrary numbers following a similar modification in Appendix A1.2.

## A1.2 EXTENSION TO MULTIPLE LAGS

**Multiple Transition Time Lag $\tau$.** For the sake of simplicity, we consider only one special case with $\tau = r + 1$ in Theorem A1. Our identifiability theorem can be actually extended to arbitrary lags directly. For any given $\tau$, according to modularity we have different conclusion at Eq 17 as **LHS** $= P(\mathbf{z}_t|\mathbf{x}_{t-1:t-\mu-r-\tau}) = P(\mathbf{z}_t|\mathbf{z}_{t-1:t-r-\tau})$. Similarity **RHS** $= P(\hat{\mathbf{z}}_t|\mathbf{x}_{t-1:t-\mu-r-\tau}) = P(\hat{\mathbf{z}}_t|\hat{\mathbf{z}}_{t-1:t-r-\tau})$ holds true as well. In addition, some modifications are needed in sufficiency assumption, i.e., re-define $\eta_{kt} \triangleq \log p(z_{kt}|\mathbf{z}_{t-1:t-r-\tau})$ and there should be at least $2n$ linear independent vectors for $\mathbf{v}, \mathring{\mathbf{v}}$ with regard to $z_{l,\eta}$ where $l = 1, 2, \cdots, n$ and $t - \tau \leq \mu \leq t - 1$. No extra changes are needed.

**Infinite Mixing Lag $r$.** The Theorem 1 can also be easily extended to infinite mixing lag since $\mathbf{z}_t = \mathbf{h}(\mathbf{z}_t; \mathbf{x}_{<t})$ still exists when $r \to \infty$, the partially invertible function.

## A1.3 NECESSITY OF CONTINUITY

Let us first give an extreme example to illustrate the importance of extra constraints for identifiability. Consider 4 independent random variables $u, v, x, y$ subjects to standard normal distribution respectively. Suppose that there exist an invertible function $(x, y) = \mathbf{h}(u, v)$ satisfies

$$\begin{cases} x = \mathbb{I}(x + y > 0) \cdot u + \mathbb{I}(x + y \leq 0) \cdot v \\ y = \mathbb{I}(x + y > 0) \cdot v + \mathbb{I}(x + y \leq 0) \cdot u. \end{cases} \quad (23)$$

Notice that the Jacobian from $(u, v)$ to $(x, y)$ contains at most one non-zero entry for each column or row. However, the result $(x, y)$ is still entangled, and the identifiability of $(u, v)$ is not achieved.

What if now we notate latent variable as $\hat{\mathbf{z}} = (u, v)$, estimated latent variable as $\mathbf{z} = (x, y)$ and the transition process with two mixing functions as $\mathbf{h} = \mathbf{g}^{-1} \circ \hat{\mathbf{g}}$?

In the literature of nonlinear ICA, the gap between $\mathbf{H}_{ij} \cdot \mathbf{H}_{ik} = 0$ when $j \neq k$ and identifiability is ill-discussed. In linear ICA, since the Jacobian is a constant matrix, these two statements are equivalent. Nevertheless, in nonlinear ICA, $\mathbf{H} = \frac{\partial \mathbf{z}}{\partial \hat{\mathbf{z}}}$ is not a constant, but a function of $\hat{\mathbf{z}}$, which may leads to the failure of identifiability as shown in Eq 23.

The counterexamples can still be easily constructed even if function $\mathbf{h}$ is continuous. For brevity, let us denote a segment-wise linear indicator function as $f(u, v) = \min(\max(0, u + v + 0.5), 1)$, and we have $\mathbf{h}$ as

$$
\begin{cases}
x = f(u, v) \cdot u + (1 - f(u, v)) \cdot v \\
y = f(u, v) \cdot v + (1 - f(u, v)) \cdot u.
\end{cases}
\tag{24}
$$

When $u, v, x, y$ are independent uniform distributions on $[-2, -1] \cup [1, 2]$, all conditions are still satisfied while the identifiability cannot be achieved.

To fill this gap, we provide two more assumptions. The domain $\hat{\mathcal{Z}}$ of $\hat{\mathbf{z}}$ should be path-connected, i.e., for any $\hat{\mathbf{z}}^{(1)}, \hat{\mathbf{z}}^{(2)} \in \hat{\mathcal{Z}}$, there exists a continuous path connecting $\hat{\mathbf{z}}^{(1)}$ and $\hat{\mathbf{z}}^{(2)}$ with all points of the path in $\hat{\mathcal{Z}}$. In addition, the derivative of function $\mathbf{h}$ is not zero for any value of $\hat{\mathbf{z}} \in \hat{\mathcal{Z}}$

**Lemma A1** (Disentanglement with Continuity). *For second-order differentiable invertible function $\mathbf{h}$ defined on a path-connected domain $\hat{\mathcal{Z}} \subseteq \mathbb{R}^n$ which satisfies $\mathbf{z} = \mathbf{h}(\hat{\mathbf{z}})$, suppose the non-degeneracy condition holds. If there exists at most one non-zero entry in each row of the Jacobian matrix $\mathbf{H} = \frac{\partial \mathbf{z}}{\partial \hat{\mathbf{z}}}$, $\hat{\mathbf{z}}$ is a disentangled version of $\mathbf{z}$ up to a permutation and a element-wise nonlinear operation.*

*Proof.* For any row $i$, $\frac{\partial \mathbf{z}_i}{\partial \hat{\mathbf{z}}} = [\frac{\partial \mathbf{z}_i}{\partial \hat{\mathbf{z}}_1}, \frac{\partial \mathbf{z}_i}{\partial \hat{\mathbf{z}}_2}, ..., \frac{\partial \mathbf{z}_i}{\partial \hat{\mathbf{z}}_n}] \in \mathbb{R}^n$ is a n-dimensional variable. Its image is a subspace as $\bigcup_{k=1}^{n} \left\{ (\frac{\partial \mathbf{z}_i}{\partial \hat{\mathbf{z}}_1}, \frac{\partial \mathbf{z}_i}{\partial \hat{\mathbf{z}}_2}, ..., \frac{\partial \mathbf{z}_i}{\partial \hat{\mathbf{z}}_n}) \in \mathbb{R}^n : \frac{\partial \mathbf{z}_i}{\partial \hat{\mathbf{z}}_j} = 0 \text{ for all } j \neq k, \text{ and } x_k \neq 0 \right\}$, since there exists at most one non-zero entry in each row of the Jacobian matrix $\mathbf{H} = \frac{\partial \mathbf{z}}{\partial \hat{\mathbf{z}}}$ and the derivative of function $\mathbf{h}$ is not zero for any value, according to the non-degeneracy condition.

We use proof by contradiction. Suppose there exist two different samples $\mathbf{a}, \mathbf{b} \in \mathcal{Z} \subseteq \mathbb{R}^n$ with different non-zero entries $j \neq k$ subjects to

$$
\left[ \frac{\partial z_i}{\partial \hat{\mathbf{z}}} \Big|_{\hat{\mathbf{z}} = \mathbf{a}} \right]_j \neq 0, \quad \left[ \frac{\partial z_i}{\partial \hat{\mathbf{z}}} \Big|_{\hat{\mathbf{z}} = \mathbf{b}} \right]_k \neq 0
\tag{25}
$$

where $[\cdot]_j$ refers to the $j$-th entry of vector. Their values are respectively within $\left\{ (0, 0, ..., \frac{\partial z_i}{\partial \hat{z}_j}, 0, ..., 0) \in \mathbb{R}^n : \frac{\partial z_i}{\partial \hat{z}_j} \neq 0 \right\}$ and $\left\{ (0, 0, ..., \frac{\partial z_i}{\partial \hat{z}_k}, 0, ..., 0) \in \mathbb{R}^n : \frac{\partial z_i}{\partial \hat{z}_k} \neq 0 \right\}$. Clearly, there is no path from $\frac{\partial z_i}{\partial \hat{\mathbf{z}}} \big|_{\hat{\mathbf{z}} = \mathbf{a}}$ to $\frac{\partial z_i}{\partial \hat{\mathbf{z}}} \big|_{\hat{\mathbf{z}} = \mathbf{b}}$. Since $\mathbf{h}$ is a second-order differentiable invertible function, we have its derivative $\mathbf{h}'$ is also differentiable. Thus, $\hat{\mathcal{Z}} \subseteq \mathbb{R}^n$ is a path-connected domain which denotes that the image of $\frac{\partial z_i}{\partial \hat{\mathbf{z}}}$ is also path-connected. It will be violated that there is no path from $\frac{\partial z_i}{\partial \hat{\mathbf{z}}} \big|_{\hat{\mathbf{z}} = \mathbf{a}}$ to $\frac{\partial z_i}{\partial \hat{\mathbf{z}}} \big|_{\hat{\mathbf{z}} = \mathbf{b}}$ thus the proof is established.

$\square$

When it comes to partially invertible function with regard to side information $\mathbf{c}$, the proof is the same with only a modification on conditions. That is, the path-connected domain assumption is applied to $(\mathbf{z}, \mathbf{c})$, and the infinite differentiability is extended to both $\mathbf{z}$ and $\mathbf{c}$, i.e., $\frac{\partial^2 z_i}{\partial a \partial b}$ for $a, b \in \{z | \mathbf{z}_i\} \times \{c | \mathbf{c}_i\}$ when $a \neq b$ exists.

Let's further review the example we provided earlier. Example in Eq 23 and Eq 24 respectively demonstrate the scenarios where the assumptions of differentiability and connectivity fail, leading to the breakdown of identifiability.

**Lemma A2** (Disentanglement with Continuity under Side Information). *For second-order differentiable invertible function $\mathbf{h}$ defined on a path-connected domain $\hat{\mathcal{Z}} \times \mathcal{C} \subseteq \mathbb{R}^{n+m}$ which satisfies $\mathbf{z} = \mathbf{h}(\hat{\mathbf{z}}, \mathbf{c})$, suppose the non-degeneracy condition holds. If there exists at most one non-zero entry in each row of the Jacobian matrix $\mathbf{H}(\mathbf{c}) = \frac{\partial \mathbf{z}}{\partial \hat{\mathbf{z}}}$, $\hat{\mathbf{z}}$ is a disentangled version of $\mathbf{z}$ up to a permutation and a element-wise nonlinear operation.*

*Proof.* Suppose there exist two different samples $\mathbf{a}, \mathbf{b} \in \hat{\mathcal{Z}} \times \mathcal{C} \subseteq \mathbb{R}^n$ with different non-zero entries $j \neq k$ subjects to

$$\left[ \frac{\partial z_i}{\partial(\hat{\mathbf{z}}, \mathbf{c})}\bigg|_{(\hat{\mathbf{z}}, \mathbf{c})=\mathbf{a}} \right]_j \neq 0, \quad \left[ \frac{\partial z_i}{\partial(\hat{\mathbf{z}}, \mathbf{c})}\bigg|_{(\hat{\mathbf{z}}, \mathbf{c})=\mathbf{b}} \right]_k \neq 0. \tag{26}$$

Similar to Lemma A1, there exists no path between them because they are blocked in $\hat{\mathcal{Z}}$ alone. In the same way, since $\mathbf{h}$ is a second-order differentiable invertible function, and the non-degeneracy condition holds, the image of $\frac{\partial z_i}{\partial(\hat{\mathbf{z}}, \mathbf{c})}$ is also path-connected. It will be violated and the proof is established.

$\square$

### A1.4 Identifiability Benefits from Non-Stationarity

We can further leverage the advantage of non-stationary data for identifiability. Let $\mathbf{v}_{kt}(u_r)$ be $\mathbf{v}_{kt}$, which is defined in Eq 11, in the $u_r$ context. Similarly, Let $\mathring{\mathbf{v}}_{kt}(u_r)$ be $\mathring{\mathbf{v}}_{kt}$ in the $u_r$ context. Let

$$\mathbf{s}_{kt} \triangleq \left( \mathbf{v}_{kt}(u_1)^\intercal, ..., \mathbf{v}_{kt}(u_m)^\intercal, \frac{\partial^2 \eta_{kt}(u_2)}{\partial z_{kt}^2} - \frac{\partial^2 \eta_{kt}(u_1)}{\partial z_{kt}^2}, ..., \frac{\partial^2 \eta_{kt}(u_m)}{\partial z_{kt}^2} - \frac{\partial^2 \eta_{kt}(u_{m-1})}{\partial z_{kt}^2} \right)^\intercal,$$

$$\mathring{\mathbf{s}}_{kt} \triangleq \left( \mathring{\mathbf{v}}_{kt}(u_1)^\intercal, ..., \mathring{\mathbf{v}}_{kt}(u_m)^\intercal, \frac{\partial \eta_{kt}(u_2)}{\partial z_{kt}} - \frac{\partial \eta_{kt}(u_1)}{\partial z_{kt}}, ..., \frac{\partial \eta_{kt}(u_m)}{\partial z_{kt}} - \frac{\partial \eta_{kt}(u_{m-1})}{\partial z_{kt}} \right)^\intercal.$$

As provided below, in our case, the identifiability of $\mathbf{z}_t$ is guaranteed by the linear independence of the whole function vectors $\mathbf{s}_{kt}$ and $\mathring{\mathbf{s}}_{kt}$, with $k = 1, 2, ..., n$. This linear independence is generally a much stronger condition.

**Corollary A1** (Identifiability under Non-Stationary Process). *Suppose $\mathbf{x}_t = \mathbf{g}(\mathbf{z}_t)$, $\mathbf{z}_t = \mathbf{m}(\mathbf{x}_{t:t-\mu})$, and that the conditional distribution $p(z_{k,t} \mid \mathbf{z}_{t-1}, \mathbf{u})$ may change across $m$ values of the context variable $\mathbf{u}$, denoted by $u_1, u_2, ..., u_m$. Suppose the components of $\mathbf{z}_t$ are mutually independent conditional on $\mathbf{z}_{t-1}$ in each context. Assume that the components of $\hat{\mathbf{z}}_t$ are also mutually independent conditional on $\hat{\mathbf{z}}_{t-1}$. Suppose the domain is path-connected and $\mathbf{m}, \hat{\mathbf{m}}, \mathbf{g}, \hat{\mathbf{g}}$ are second-order differentiable and their combination subjects to non-degenerate condition. If the $2n$ function vectors $\mathbf{s}_{k,t}$ and $\mathring{\mathbf{s}}_{k,t}$, with $k = 1, 2, ..., n$, are linearly independent, then $\hat{\mathbf{z}}_t$ is a permuted invertible component-wise transformation of $\mathbf{z}_t$.*

*Proof.* Drawing upon the arguments in the proof of Theorem 1, given that the components of $\hat{\mathbf{z}}_t$ are mutually independent conditional on $\hat{\mathbf{z}}_{t-1}$, we know that for $i \neq j$,

$$\frac{\partial^2 \log p(\hat{\mathbf{z}}_t \mid \hat{\mathbf{z}}_{t-1}; \mathbf{u})}{\partial \hat{z}_{it} \partial \hat{z}_{jt}} = \sum_{k=1}^{n} \left( \frac{\partial^2 \eta_{kt}(\mathbf{u})}{\partial z_{kt}^2} \cdot \mathbf{H}_{kit} \mathbf{H}_{kjt} + \frac{\partial \eta_{kt}(\mathbf{u})}{\partial z_{kt}} \cdot \frac{\partial \mathbf{H}_{kit}}{\partial \hat{z}_{jt}} \right) - \frac{\partial^2 \log |\mathbf{H}_t|}{\partial \hat{z}_{it} \partial \hat{z}_{jt}} \equiv 0. \tag{27}$$

In contrast to Eq 21, we now allow $p(\hat{\mathbf{z}}_t \mid \hat{\mathbf{z}}_{t-1})$ to depend on $\mathbf{u}$. Given that the aforementioned equation is always 0, its partial derivative w.r.t. $z_{l,t-1}$ yields

$$\frac{\partial^3 \log p(\hat{\mathbf{z}}_t \mid \hat{\mathbf{z}}_{t-1}; \mathbf{u})}{\partial \hat{z}_{it} \partial \hat{z}_{jt} \partial z_{l,t-1}} = \sum_{k=1}^{n} \left( \frac{\partial^3 \eta_{kt}(\mathbf{u})}{\partial z_{kt}^2 \partial z_{l,t-1}} \cdot \mathbf{H}_{kit} \mathbf{H}_{kjt} + \frac{\partial^2 \eta_{kt}(\mathbf{u})}{\partial z_{kt} \partial z_{l,t-1}} \cdot \frac{\partial \mathbf{H}_{kit}}{\partial \hat{z}_{jt}} \right) \equiv 0. \tag{28}$$

Similarly, when using varied values for $\mathbf{u}$ in Eq 27, computing the difference between these instances yields

$$
\frac{\partial^2 \log p(\hat{\mathbf{z}}_t \mid \hat{\mathbf{z}}_{t-1}; u_{r+1})}{\partial \hat{z}_{it} \partial \hat{z}_{jt}} - \frac{\partial^2 \log p(\hat{\mathbf{z}}_t \mid \hat{\mathbf{z}}_{t-1}; u_{r+1})}{\partial \hat{z}_{it} \partial \hat{z}_{jt}}
$$
$$
= \sum_{k=1}^{n} \left[ \left( \frac{\partial^2 \eta_{kt}(u_{r+1})}{\partial z_{kt}^2} - \frac{\partial^2 \eta_{kt}(u_r)}{\partial z_{kt}^2} \right) \cdot \mathbf{H}_{kit} \mathbf{H}_{kjt} + \left( \frac{\partial \eta_{kt}(u_{r+1})}{\partial z_{kt}} - \frac{\partial \eta_{kt}(u_r)}{\partial z_{kt}} \right) \cdot \frac{\partial \mathbf{H}_{kit}}{\partial \hat{z}_{jt}} \right] \equiv 0.
\tag{29}
$$

Therefore, if $\mathbf{s}_{kt}$ and $\mathring{\mathbf{s}}_{kt}$, for $k = 1, 2, ..., n$, are linearly independent, $\mathbf{H}_{kit} \mathbf{H}_{kjt}$ has to be zero for all $k$ and $i \neq j$. Building on the insights from the proof of Theorem 1, $\hat{\mathbf{z}}_t$ is compelled to be a permutation of a component-wise invertible transformation of $\mathbf{z}_t$. $\qquad\square$

## A2 EXPERIMENT SETTINGS

### A2.1 REPRODUCIBILITY

All experiments are done in a GPU workstation with CPU: Intel(R) Xeon(R) Platinum 8168 CPU @ 2.70GHz, GPU: Tesla V100. The source code and the generated data for the simulation experiments are attached in the supplementary materials.

### A2.2 SYNTHETIC DATASET GENERATION

In this section, we give 2 representative simulation settings for **NG** and **NG-TDMP** respectively to reveal the identifiability results. For each synthetic dataset, we set latent space to be 3, i.e., $\mathbf{x}_t \in \mathcal{X} \subseteq \mathbb{R}^3$.

**Non-invertible Generation** For **NG**, we set the transition lag as $\tau = 1$. We first generate $10,000$ data points from uniform distribution as the initial state $\mathbf{z}_0 \sim U(0,1)$. For $t = 1, \cdots, 9$, each latent variable $\mathbf{z}_t$ will be generated from the proceeding latent variable $\mathbf{z}_{t-1}$ through a nonlinear function $\mathbf{f}$ with a non-additive zero-biased Gaussian noise $\epsilon_t$ ($\sigma = 0.1$), i.e., $\mathbf{z}_t = \mathbf{f}(\mathbf{z}_t, \epsilon_t)$. To introduce the non-invertibility, the mixing function $\mathbf{g}$ leverages only the first two entries of the latent variables to generate the 2-d observation $\mathbf{z}_t = \mathbf{g}(x_{1,t}, x_{2,t}) \in \mathcal{Z} \subseteq \mathbb{R}^2$.

**Time-Delayed Mixing Process** For **UG-TDMP**, we set the transition lag as $\tau = 1$ and mixing lag $r = 2$. Similar to the Non-invertible Generation scenario, we generate the initial states from uniform distribution and the subsequent latent variables following a nonlinear transition function. The noise is also introduced in a nonlinear Gaussian ($\sigma = 0.1$) way. The mixing process is a nonlinear function with regard to $\mathbf{z}_t$ plus a side information from previous steps $\mathbf{z}_{t-1:t-2}$, i.e.,

$$
\mathbf{x}_t = A_{3\times3} \cdot \sigma\big(B_{3\times3} \cdot \sigma(C_{3\times3} \cdot \mathbf{z}_t)\big) + \begin{bmatrix} 0 \\ 0 \\ D_{3\times1} \mathbf{z}_{t-1} + E_{3\times1} \mathbf{z}_{t-2} \end{bmatrix},
\tag{30}
$$

where $\sigma$ refers to the ReLU function and the capital characters refer to matrices. Note that we make two modifications to show the advantage of `CaRiNG`. The reason we consider larger mixing lag is that it is a much more difficult scenario to handle, with more distribution from the mixing process and less dynamic information from transition. We run experiments in both scenarios with different transition and mixing lag. Besides, we also find out that even without time-lagged latent variables in the decoder, it leads to a smaller model that is more stable and easy to train. Refer to Table A1 for a detailed ablation study.

**Post-processing Precedure** During the generating process, we did not explicitly enforce the data to meet the constraint $\mathbf{z}_t = \mathbf{m}(\mathbf{x}_{t:t-\mu})$. On the contrary, we implement a checker to filter the data that is qualified. To be more precise, we do linear regression from $\mathbf{x}_{t:t-\mu}$ to $\mathbf{z}_t$ to figure out how much information of latent variables can be recovered from observation series in the best case. We choose the smallest $\mu$ when the amount of information that can be recovered is acceptable. We set $\mu = 2$ for **UG** and $\mu = 4$ for **UG-TDMP**.

| setting | $\tau = 1, r = 2$ | $\tau = 2, r = 1$ |
|---|---|---|
| **CaRiNG** | 0.9436 | 0.9131 |
| **CaRiNG** (lagged decoder) | 0.9250 | 0.9220 |
| TDRL | 0.8947 | 0.7519 |

Table A1: Ablation study on different settings for **UG-TDMP**. (a) The second column is a more difficult scenario compared to the first, where the performance of **CaRiNG** remains good while that of baseline decreases significantly. (b) Omit the time-lagged latent variables in the decoder will not damage the performance much, but one can enjoy the benefits from a much simpler model.

### A2.3 IMPLEMENTATION DETAILS

#### A2.3.1 SYNTHETIC DATA

**Network Architecture** To implement the Sequence-to-Step encoder, we leverage the *torch.unfold* to generate the nesting observations. Let us denote $\mathbf{x}_t^{(\mu)} = [\mathbf{x}_t, \cdots, \mathbf{x}_{t-\mu}]$ as inputs. For the time steps that do not exist, we simply pad them with zero. Refer to Table A2 for detailed network architecture.

**Training Details** The models were implemented in PyTorch 1.11.0. An AdamW optimizer is used for training this network. We set the learning rate as $0.001$ and the mini-batch size as $64$. We train each model under four random seeds ($770, 771, 772, 773$) and report the overall performance with mean and standard deviation across different random seeds.

Table A2: Architecture details. BS: batch size, T: length of time series, i_dim: input dimension, o_dim: output dimension, z_dim: latent dimension, LeakyReLU: Leaky Rectified Linear Unit.

| Configuration | Description | Output |
|---|---|---|
| **1. Sequence-to-Step Encoder** | Encoder for Synthetic Data | |
| Input: $\mathbf{x}_{1:T}^{(\mu)}$ | Observed time series | BS × T × i_dim |
| Dense | 128 neurons, LeakyReLU | BS × T × 128 |
| Dense | 128 neurons, LeakyReLU | BS × T × 128 |
| Dense | 128 neurons, LeakyReLU | BS × T × 128 |
| Dense | Temporal embeddings | BS × T × z_dim |
| **2. Step-to-Step Decoder** | Decoder for Synthetic Data | |
| Input: $\hat{\mathbf{z}}_{1:T}$ | Sampled latent variables | BS × T × z_dim |
| Dense | 128 neurons, LeakyReLU | BS × T × 128 |
| Dense | 128 neurons, LeakyReLU | BS × T × 128 |
| Dense | i_dim neurons, reconstructed $\hat{\mathbf{x}}_{1:T}$ | BS × T × o_dim |
| **3. Factorized Inference Network** | Bidirectional Inference Network | |
| Input | Sequential embeddings | BS × T × z_dim |
| Bottleneck | Compute mean and variance of posterior | $\mu_{1:T}, \sigma_{1:T}$ |
| Reparameterization | Sequential sampling | $\hat{\mathbf{z}}_{1:T}$ |
| **4. Modular Prior** | Nonlinear Transition Prior Network | |
| Input | Sampled latent variable sequence $\hat{\mathbf{z}}_{1:T}$ | BS × T × z_dim |
| InverseTransition | Compute estimated residuals $\hat{\epsilon}_{it}$ | BS × T × z_dim |
| JacobianCompute | Compute $\log(|\det(\mathbf{J})|)$ | BS |

#### A2.3.2 REAL-WORLD DATASET

**Network Architecture** We choose HCRN (Le et al., 2020) (without classification head) as the encoder backbone of **CaRiNG** on the real-world dataset: SUTD-TrafficQA. Given that HCRN is

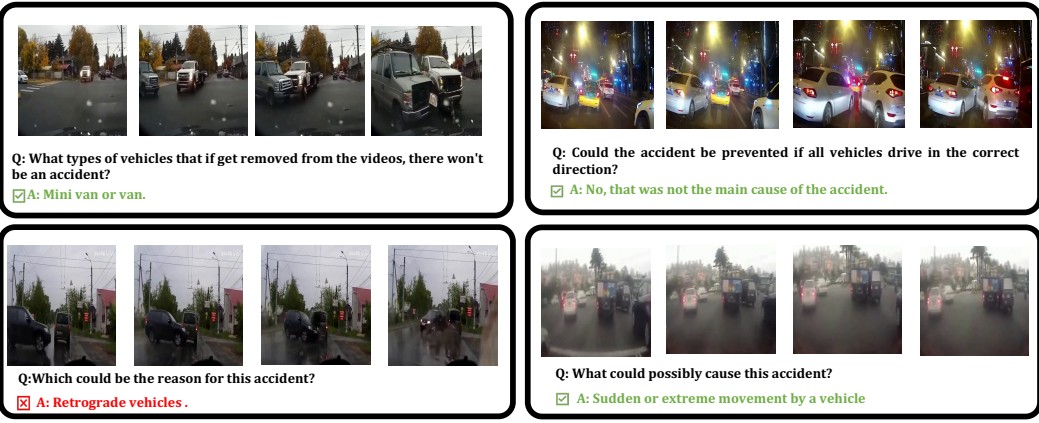

Figure A1: **Qualitative resutls on SUTD-TrafficQA dataset.** We provide some positive examples and also fail cases to analyze our model.

an encoder that calculates the cross attention between visual input and text input sequentially, we apply a decoder, which shares the same structure as the Step-to-Step Decoder shown in Table A2 to reconstruct the visual feature embedded with the temporal information. As it goes to transition prior, we use the Modular Prior shown in Table A2. This encoder-decoder structure can guide the model to learn the hidden representation with identifiable guarantees under the non-invertible generation process.

## A3 MORE VISUALIZATION RESULTS ON REAL-WORLD DATA

As shown Table A1, we provide some positive examples and also fail cases to analyze our model. From the top two examples, we can find that our method can solve the occlusions well. From the bottom right one, we find that our model can solve the blurred situation. However, when the alignment between visual and textual domains is difficult. The model may fail.

## A4 MORE EXPERIMENTAL RESULTS

### A4.1 COMPUTATION COST COMPARISON

We provide the comparisons between the computational cost of the `CaRiNG` model compared to HCRN to analyze our efficiency. As shown in Table A3, we provide a detailed comparison of the number of parameters, training time, and inference efficiency. It is important to note that while the CaRiNG model requires a longer training time due to the application of normalizing flow for calculating the Jacobian matrix, its inference efficiency remains on par with HCRN, as the normalizing flow is utilized only for calculating KL loss and not during inference.

| Method | HCRN | CaRiNG |
|---|---|---|
| Number of Parameters | 42,278,786 | 43,721,954 |
| Training Time per Epoch | 6min 54s/epoch | 13min 26s/epoch |
| Inference Time per Epoch | 49s/epoch | 49s/epoch |

Table A3: Comparative Analysis of HCRN and `CaRiNG` Models

This analysis clearly demonstrates that the increased training time for the CaRiNG model is offset by its comparable inference efficiency, highlighting its practical applicability in scenarios where inference time is critical.

### A4.2 EVALUATION OF IDENTIFIABILITY IN THE QA BENCHMARK

In the context of real-world applications, particularly in scenarios lacking ground truth for rigorous metrics like MCC, alternative evaluation strategies become essential. we leverage proxy metrics to assess the performance of the proposed algorithm, focusing on two pivotal aspects: disentanglement and reconstruction ability of the learned representations. Intuitively, as delineated in Theorem A1 and detailed in Section 4, a representation can be considered identifiable if it possesses the dual capability of fully reconstructing the observation while also achieving disentanglement. Thus, as a supplement to the accuracy we used before, we benchmark disentanglement and reconstruction ability as side evidence to support that the improvement is caused by better identifiability.

We use the ELBO loss as a proxy metric to evaluate the identifiability. Figure A2 illustrates our method's performance compared to the baseline TDRL method. The results clearly show that our approach exhibits superior disentanglement and reconstruction abilities. This evidence suggests that the advantage of our proposed algorithm is primarily attributed to its enhanced identifiability and effective disentanglement of data representations.

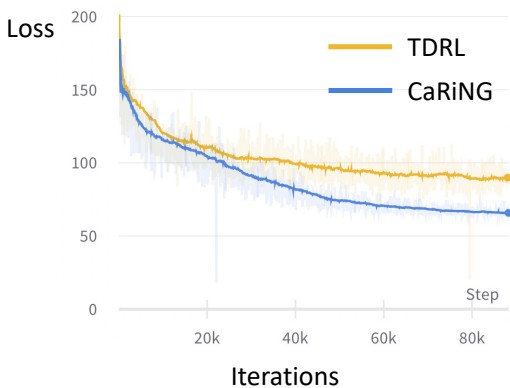

Figure A2: Comparative analysis of disentanglement and reconstruction abilities of different methods.

### A4.3 PARAMETER ANALYSIS ON $\tau$ IN SUTD-TRAFFICQA

In this section, we present the results of our parameter analysis conducted on the SUTD-TrafficQA dataset, focusing on the impact of varying the time lag $\tau$. The study aimed to assess the robustness of our model to changes in the time lag parameter. As the table below illustrates, the model demonstrates consistent accuracy across different values of $\tau$, indicating robustness to the variation in time lag.

| $\tau$ | 1 | 2 | 3 |
|---|---|---|---|
| Accuracy (%) | 41.22 | 41.23 | 41.27 |

Table A4: Parameter analysis results of $\tau$ on model accuracy in the SUTD-TrafficQA dataset.

### A4.4 MODEL SELECTION WITH VARYING $\mu$

In this subsection, we discuss a preliminary experiment that was instrumental in the model selection process for our application in the NG-TDMP settings. The experiment focused on evaluating the performance of the model with varying lengths of time lag $\mu$.

Our findings indicate that an increase in $\mu$ does not always correlate with enhanced model performance. We observed that the effectiveness of each latent variable diminishes as the time lag $\mu$ increases. In

practical applications, this motivates a strategy of model selection where an appropriate value of $\mu$ is chosen based on the model's performance. The following table summarizes our experimental results:

| $\mu$ | 3 | 4 | 5 |
|---|---|---|---|
| Accuracy (%) | 0.88 | 0.92 | 0.92 |

Table A5: Impact of varying $\mu$ on model performance in NG-TDMP settings.

These results suggest that while a larger $\mu$ might imply a more extensive recovery of context information, it can also introduce inefficiencies in information recovery, potentially adding noise and impeding model training.

## A5 RELATED WORK

### A5.1 CAUSAL DISCOVERY WITH LATENT VARIABLES

Some studies have aimed to discover causally related latent variables, such as Silva et al. (2006); Kummerfeld & Ramsey (2016); Huang et al. (2022) leverage the vanishing Tetrad conditions Spearman (1928) or rank constraints to identify latent variables in linear-Gaussian models, and Shimizu et al. (2009); Cai & Xie (2019); Xie et al. (2020; 2022) draw upon non-Gaussianity in their analysis for linear, non-Gaussian scenarios. Furthermore, some methods aim to find the structure beyond the latent variables, resulting in the hierarchical structure. Some hierarchical model-based approaches assume tree-like configurations, such as Pearl (1988); Zhang (2004); Choi et al. (2011); Drton et al. (2017), while the other methods assume a broader hierarchical structure Xie et al. (2022); Huang et al. (2022). However, these methods remain confined to linear frameworks and face escalating challenges with intricate datasets, such as videos.

### A5.2 NONLINEAR ICA FOR TIME SERIES DATA

Nonlinear ICA represents an alternative methodology to identify latent causal variables within time series data. Such methods leverage auxiliary data—like class labels and domain indices—and impose independence constraints to facilitate the identifiability of latent variables. To illustrate: Time-contrastive learning (TCL (Hyvarinen & Morioka, 2016)) adopts the independent sources premise and capitalizes on the variability in variance across different data segments. Furthermore, Permutation-based contrastive (PCL (Hyvarinen & Morioka, 2017)) puts forth a learning paradigm that distinguishes genuine independent sources from their permuted counterparts. Furthermore, i-VAE (Khemakhem et al., 2020) utilizes deep neural networks, VAEs, to closely approximate the joint distribution encompassing observed and auxiliary non-stationary regimes. Recent work, exemplified by LEAP (Yao et al., 2022b), has tackled both stationary and non-stationary scenarios in tandem. In the stationary context, LEAP postulates a linear non-Gaussian generative process. For the non-stationary context, it assumes a nonlinear generative process, gaining leverage from auxiliary variables. Advancing beyond LEAP, TDRL (Yao et al., 2022a) initially extends the linear non-Gaussian generative assumption to a nonlinear formulation for stationary scenarios. Subsequently, it broadens the non-stationary framework to accommodate structural shifts, global alterations, and combinations thereof. Additionally, CITRIS (Lippe et al., 2022b;a) champions the use of intervention target data to precisely identify scalar and multi-dimensional latent causal factors. However, a common thread across these methodologies is the presumption of an invertible generative process, a stance that often deviates from the realities of actual data.

## A6 MORE DISCUSSIONS

To make our contribution more clear, we provide more discussions with other related work.

### A6.1 COMPARISON TO (LACHAPELLE ET AL., 2023)

Concurrently to our work, Lachapelle et al. (2023) rightfully points out that one has to be careful when going from "Jacobian is a permutation-scaling matrix" to "the mapping is a permutation composed with an element-wise transformation" when the domain of the function is not simply $\mathbf{R}^n$. This conclusion strongly supports our viewpoint. In their context, the terms "local disentanglement" and "global disentanglement" are used.

### A6.2 COMPARISON WITH SEQUENTIALVAE CHUNG ET AL. (2015)

In this subsection, we provide a detailed exploration of the unique aspects of our approach, CaRiNG, distinguishing it from the original Sequential VAE:

- **Identifiability Theory:** Our approach extends the identifiability theory to scenarios involving non-invertible mixing functions, enhancing the existing ICA framework. We introduce theoretical constructs that enable the identification of latent variables even when the mixing function is non-invertible. This theory paves the way for a more robust and accurate extraction of latent factors from mixed signals.

- **Temporal Dynamics Modeling: `CaRiNG`**incorporates a transition function for capturing the temporal dynamics of multivariate data. By integrating such a transition function, our model can effectively track and represent the causal relations of latent variables over time. This aspect is particularly vital in understanding and predicting time-series data where the temporal relationship plays a crucial role.

- **Prior Module for Conditional Independence:** A novel aspect of `CaRiNG`is the introduction of a prior module specifically designed to enforce the conditional independence of latent variables. This module aids in disentangling the latent space by ensuring that the dependencies among latent variables are captured more explicitly and accurately. By promoting conditional independence, our model enhances the clarity and interpretability of the latent representations, which is a significant step forward in latent variable modeling.

These enhancements position CaRiNG as a method focused on learning causal representations with clear identifiability guarantees, marking a departure from the generation-centric objectives commonly seen in traditional VAE-based methods. Our method's ability to provide clearer, more interpretable latent representations makes it particularly valuable in complex data analysis and modeling scenarios.

Experimentally, we also compared the identifiability of the learned variables between our methods and other SequentialVAE-based methods, including SKD Berman et al. (2022) and SequentialVAE Chung et al. (2015). As shown in Table 1 and Figure 4, our method shows significant improvement over SKD. Interestingly, we find that the SequentialVAE method works better than other methods that don't use the temporal context, which also demonstrates the necessity of temporal context to solve the invertibility issue. However, we still find that constraining conditional independence benefits better performance.

### A6.3 COMPARISON TO OTHER METHODS USING NORMALIZING FLOW REZENDE & MOHAMED (2015); ZIEGLER & RUSH (2019)

Our approach, while applying normalizing flows Rezende & Mohamed (2015), is distinct from existing works in its motivation and implementation:

- **Application of Normalizing Flows for Prior Distribution:** In CaRiNG, normalizing flows are uniquely utilized for the computation of the prior distribution, which is a fundamental aspect of our model's architecture. It plays a vital role in constraining the conditional independence of latent variables. For existing works Rezende & Mohamed (2015); Ziegler

& Rush (2019), they apply the normalizing flow to obtain the invertible deterministic transformation between two variables in the sequence, which is distinct from ours.

- **Sequence-to-Step Encoder:** Our model incorporates a sequence-to-step encoder, specifically designed to leverage the temporal context of data. This encoder is adept at addressing the challenges brought about by non-invertible mixing functions. Unlike existing methods that predominantly focus on current observations, our encoder takes into account the temporal context of data.

By integrating these unique features, `CaRiNG`leans the disentangled latent variables with an identifiability guarantee. This methodological distinction underscores our contribution to the field of VAE models, even though we all apply normalizing flows as a tool.

## A7   BROADER IMPACTS, LIMITATION, AND FUTURE WORK

This study introduces both a theoretical framework and a practical approach for extracting causal representations from time-series data. Such advancements enable the development of more transparent and interpretative models, enhancing our grasp of causal dynamics in real-world settings. This approach may benefit many real-world applications, including healthcare, auto-driving, and finance, but it could also be used illegally. For example, within the financial sphere, it can be harnessed to decipher ever-evolving market trends, optimizing predictions and thereby influencing investment and risk management decisions. However, it's imperative to note that any misjudgment of causal relationships could lead to detrimental consequences in these domains. Thus, establishing causal links must be executed with precision to prevent skewed or biased inferences.

Theoretically, though allowing for the non-invertible generation process, our theoretical assumptions still fall short of fully capturing the intricacies of real-world scenarios. For example, identifiability requires the absence of instantaneous causal relations, i.e., relying solely on time-delayed influences within the latent causal dynamics. Furthermore, we operate under the presumption that the number of variables remains consistent across different time steps, signifying that no agents enter or exit the environment. Moving forward, we aim to broaden our framework to ensure identifiability in more general settings, embracing instantaneous causal dynamics and the flexibility for variables to either enter or exit.

In our experiments, we evaluate our approach with both simulated and real-world datasets. However, our simulation relies predominantly on data points, creating a gap from real-world data. Concurrently, the real datasets lack the presence of ground truth latent variables. In the future, we plan to develop a benchmark specifically tailored for the causal representation learning task. This benchmark will harness the capabilities of game engines and renderers to produce videos embedded with ground-truth latent variables.

