# OpenReview forum: "Learning Temporal Causal Representation under Non-Invertible Generation Process"
_ICLR.cc/2024/Conference — Submitted to ICLR 2024_

### Official Review · Reviewer_uMXb · 2023-10-30

**Soundness:** 3 good
**Presentation:** 3 good
**Contribution:** 3 good
**Rating:** 8
**Confidence:** 2

**Summary:**

The authors propose a model recovering latent causal factors of a time-series, in other words inverting the generating process of sequential data. The key feature of this model is that it utilises temporal context (i.e. to recover latent factors at time t, it uses observations at time t, t-1, ..., t - k for some k) which allows us to overcome the non-injectivity of the generating function. The model is motivated by a theoretical analysis showing that under certain assumptions such an inversion is guaranteed to recover the true latent factors. The numerical experiments demonstrate superior performance of the proposed model in comparison to a number of baselines on synthetic dataset and real-world datasets.

**Strengths:**

+ An interesting model addressing important questions of nonlinear identifiability and disentanglement in temporal data
+ Thorough theoretical analysis of the proposed model
+ Experimental comparisons to a number of baseline models

**Weaknesses:**

I think the presentation could be somewhat improved. The paper is full of technical details of the identifiability theory but I think it would also benefit from a higher-level discussion (maybe using a cartoon or a toy-example) illustrating the intuition behind the assumptions of the theorems. My take home message after reading this paper is that using temporal context enables nonlinear identifiability and disentanglement under certain conditions, but I'd be struggling to explain what these conditions mean and why the temporal context is so crucial.

**Questions:**

Questions to Definition 1:
- Why are m and \hat{m} in the subscript of distributions in Eq. (3)? As I understood m is not part of the generative model (we don't need it to generate data from the latent factors) so it shouldn't influence the resulting data distribution.
- Should the model and data distributions match almost everywhere rather than everywhere?
- According to this definition, the latent process is identifiable if the model and data distributions match and \hat{m} = m up to permutations. What about the case when the model and data distributions don't match but \hat{m} = m up to permutations? (for example, if we could obtain true m-function with the wrong model) Is it an impossible scenario or rather just not in the scope of this paper?

Questions to Theorem 1:
- Is it possible to estimate how much temporal context (i.e. the value of \mu) is required for identifiability in Theorem 1? Or does the result only say that if the inverting function m exists for some \mu then we can estimate it up to permutations but we don't know how much temporal history we might need to that?
- (A more speculative question, feel free to ignore if it doesn't make sense.) Do you have an intuition what happens as \mu -> \infty? Is every model identifiable in the limit or not necessarily?

Question to Section 4:
- "To enforce the conditional independence of latent variables, the distribution of p(z_t | x_{t:t−μ}) is constrained by the prior." Why does the prior constrain the conditional independencies in the posterior? I guess you refer to ELBO which includes a KL divergence to the prior, but the global maximiser of ELBO is the true posterior distribution, and clearly there are examples of models with independent prior but dependent posterior (e.g. https://en.wikipedia.org/wiki/Interaction_information#Negative_interaction_information)

---

> ### Author Response · Authors · 2023-11-18
> **Response to Reviewer uMXb Part 1**
>
> Dear Reviewer uMXb, we would like to express our deep gratitude for your kind approval and valuable suggestions provided on our paper, as well as for the time you devoted to reviewing it. Below, we have addressed each of your comments and suggestions.
>
> > **W1:** I think the presentation could be somewhat improved. The paper is full of technical details of the identifiability theory but I think it would also benefit from a higher-level discussion (maybe using a cartoon or a toy-example) illustrating the intuition behind the assumptions of the theorems. My take home message after reading this paper is that using temporal context enables nonlinear identifiability and disentanglement under certain conditions, but I'd be struggling to explain what these conditions mean and why the temporal context is so crucial.
>
>
> **A1:** We are grateful for your valuable suggestions which helped better explain our insights.  The identifiability results in our theorem rely on the assumption that there exists a function $\mathbf{m}$ to recover the information lost caused by non-invertibility from the temporal context.
> The intuitive insights leveraging the temporal context come from the human visual perception system.  We illustrate our concept with two intuitive examples.
>
> **The perception of 3D world using vidoe**
> The process of visual perception can be understood as a non-invertible transformation, where a three-dimensional world is projected onto two-dimensional images. This transformation inherently loses some spatial information due to its non-invertible nature. However, by analyzing multiple views of the same object in a video, we can compensate for this loss. Each view offers a unique perspective, providing complementary information that collectively helps in reconstructing a more complete understanding of the object in its original three-dimensional form.
>
>
> **Inference with occlusion**
> Similarly, consider an example about the inference of the location of a ball that is totally occluded. The occlusion is a non-invertible function. Our humans often have the ability to infer the location of the occluded ball, since we can obtain the information from temporal contexts.
>
>
> To elaborate on why function $\mathbf{m}$ exists and what these conditions mean, we propose both an intuitive example and a mathematical example (with a carton) using visual persistence. Please refer to Section 2.3 and Figure 2 in the revised paper for more details.
>
> > **W2(Q1):** Why are m and \hat{m} in the subscript of distributions in Eq. (3)? As I understood m is not part of the generative model (we don't need it to generate data from the latent factors) so it shouldn't influence the resulting data distribution.
>
> **A2:** We greatly appreciate your suggestions. Although $\mathbf{m}$ does not explicitly participate in the generation process, it does so implicitly as a property of the mixing function $\mathbf{g}$. That is why we put $\mathbf{m}$ into the definition part in the manuscript. However, we do agree that this may cause confusion. Therefore, we have removed $\mathbf{m}$ from the definition part and added an explanation in the paragraph below Definition 1.
>
> > **W3(Q2):** Should the model and data distributions match almost everywhere rather than everywhere?
>
> **A3:**  Thanks a lot for pointing out this question. By 'everywhere' we would like to say that 'for any value of $\mathbf{x}_t$'. We have corrected this issue in Definition 1 in the revised version.
>
> > **W4(Q3):** According to this definition, the latent process is identifiable if the model and data distributions match and \hat{m} = m up to permutations. What about the case when the model and data distributions don't match but \hat{m} = m up to permutations? (for example, if we could obtain true m-function with the wrong model) Is it an impossible scenario or rather just not in the scope of this paper?
>
> **A4:**   Thanks a lot for your valuable questions. Actually, the aim here is to figure out under what conditions can the process be identifiable. It is totally possible that we get the true process, up to a permutation and element-wise nonlinear function, while the data distributions don't match. Take a example, we have $\hat{z} = z$ and $\hat{m} = \pi \circ m$. Both $\hat{z},\hat{m}$ are versions of permutated truth. However, since the permutations are not matched, the data distributions don't match.  To confirm, it is not in the scope of this paper.

---

> ### Author Response · Authors · 2023-11-18
> **Response to Reviewer uMXb Part 2**
>
> > **W6(Q5):** (A more speculative question, feel free to ignore if it doesn't make sense.) Do you have an intuition what happens as \mu -> \infty? Is every model identifiable in the limit or not necessarily?
>
> **A6:**  Great insights! Thanks for this question. In our proof, for any given $\mu$ it will work whether it is infinite or not. Actually, when $\mu$ is infinite, the temporal context involves more information ideally. We provided an explicit function of $m$ in a toy example if $\mu$ is infinite.
>
> **Notice:** Due to the issue of markdown for complex equations, we use   $a\underline{ }(b)$ to represent $a_b$.
>
> Let $\mathbf{x}\underline{ }t = \mathbf{g}(\mathbf{z}\underline{ }{(t:t-1)}) = \frac{2}{3}\mathbf{z}\underline{ }t + \frac{1}{3}\mathbf{z}\underline{ }{(t-1)}.$
> Given an observed sequence with length $\mu$, the current latent variable can be rewritten as $\mathbf{z}\underline{ }t  = \frac{3}{2}\mathbf{x}\underline{ }t - \frac{1}{2}\mathbf{z}\underline{ }{(t-1)} = \left(-\frac{1}{2}\right)^{\mu+1}\mathbf{z}\underline{ }{(t-\mu-1)} + \frac{3}{2}\sum_{i=0}^\mu \left(-\frac{1}{2}\right)^{i}\mathbf{x}\underline{ }{(t-i)}.$
> Thereby we can constructing a function $\mathbf{m}$ as $\mathbf{m}(\mathbf{x}\underline{ }{(t:t-\mu)})=\frac{3}{2}\sum_{i=0}^\mu \left(-\frac{1}{2}\right)^{i}\mathbf{x}\underline{ }{(t-i)}$. When we have an infinite observation sequence, we have $\lim_{\mu\rightarrow\infty}\mathbf{m}(\mathbf{x}\underline{ }{(t:t-\mu)}) = \mathbf{z}\underline{ }t$.
>
> In practice, an increasing $\mu$ will bring more noise, which may affect the model training. So we apply the model selection to determine $\mu$ in practice.
> Please refer to Appendix A4.4 in the revised paper for more details.
>
> > **W7(Q6):** "To enforce the conditional independence of latent variables, the distribution of p(z\underline{ }t | x\underline{ }{t:t−μ}) is constrained by the prior." Why does the prior constrain the conditional independencies in the posterior? I guess you refer to ELBO which includes a KL divergence to the prior, but the global maximiser of ELBO is the true posterior distribution, and clearly there are examples of models with independent prior but dependent posterior (e.g. https://en.wikipedia.org/wiki/Interaction\underline{ }information#Negative\underline{ }interaction\underline{ }information)
>
> **A7:** We respectfully appreciate your insightful question. Sorry for the inconvenience caused by the unclear writing.
>
> When we say "To uphold the conditional independence assumption", we didn't mean that the KL divergence enforces entries of $z\underline{ }t$ to be conditional independent on $\mathbf{\hat{z}}\underline{ }{(t-1:t-\tau)}$ with each other. Instead, what we want to say is that by minimizing the KL divergence between prior $p(\mathbf{\hat{z}}\underline{ }t|\mathbf{\hat{z}}\underline{ }{(t:t-\tau)})$ and posterior $q(\mathbf{\hat{z}}\underline{ }t|\mathbf{x}\underline{ }{(t:t-\mu)})$, the posterior is also conditionally independent on $\mathbf{\hat{z}}\underline{ }{(t-1:t-\tau)}$, since the conditional independence is hard-coded in the calculation of prior.
> We reorganized the paragraph "Transition Prior Module" in Section 4 to make it more clear.
>
> After all, many thanks for your effort to enhance the readability of our paper.

---

> > ### Comment · Reviewer_uMXb · 2023-11-20
> >
> > Thank you for your detailed reply! I don't have further questions at this stage.

---

> ### Author Response · Authors · 2023-11-21
> **Thanks for your prompt feedback**
>
> Dear Reviewer uMXb,
>
> We are pleased to hear that your issues are totoally addressed. Thank you for your support and recognition of our work.
>
> Best regards,
>
> Authors of submission 2088

---

### Official Review · Reviewer_ur5z · 2023-10-30

**Soundness:** 2 fair
**Presentation:** 1 poor
**Contribution:** 3 good
**Rating:** 3
**Confidence:** 4

**Summary:**

The authors propose a novel approach for latent variable identification in a temporal setting. It relaxes the common assumption that the latent representation at time step t can be uniquely determined from the observation at that time step. Instead it assumes that it can be determined from a window of past observations. Their results rely on sufficient variability assumptions very similar to what has been proposed in the literature. They proposed an algorithm based of VAEs and normalizing flows to model the transition model in the latent space. They show experiments on synthetic data, to validate identifiability, and on realistic QA datasets, to assess the usefulness of the learned representation.

### **Review summary**
Although I believe the motivation and the proposition of this paper is very good, I believe this manuscript is not ready for publication. My main concerns are:
- I am not certain that the there exists a model that actually satisfy the assumptions of this work.
- The paper presents many math mistakes and imprecision. The terminology used is also wrong at times.
- The point made in Section 3.3 was already made in a previous work [4]. This work also presents a counter example to Lemma 1 (implying that it is false).
- The writing quality is low
- The paper is not well situated in the literature, which makes it hard for a non-expert to understand what is the actual novelty.

I substantiate all of these points below, in the "Weakness" section. I sincerely believe this idea has great potential, but too many problems in the execution. For these reasons, I recommend rejection. I very much hope that the authors will take my criticisms seriously and use them to improve their work.

### **Post discussion phase**

Looks like the discussion phase is over. I was hoping to answer the last points raised by the authors, but couldn't do it in the comments, so I decided to share it here:

----

Concrete mathematical example: Well, if you take $f_i$ to be noisy here, what is the corresponding $m$? You provided an $m$
only for the non-noisy case, but your theory assumes noise. And my guess is that the noise is crucial to your proofs (as is often the case in nonlinear ICA).

Counter-example to Lemma 1: Indeed, if you change your definition of disentanglement to having the "same permutation everywhere" then the result is correct. But the current phrasing of the Lemma does not suggest this definition, so Example 6 is indeed a counter-example. It's impossible for me to review your modification and make sure it is correct.

This is a very mathematical paper. I feel like many non-trivial changes to the paper have been done. For instance, the whole section on the model definition has been updated. For some reason, I cannot view the previous versions of the paper, but iirc the mixing function use to take as input z_t and output x_t, correct? This seems to be corroborated by Figure 3 where the StepDecoder has as input only z_t. In the current revision, the model definition allows g to take as input a window of past z_t. This is a non-trivial change in my opinion. What are the repercussions of this change to your proof and the rest of the paper? This is only one of the many changes that this manuscript received. IMO, this version requires a full rereading to make sure everything adds up, i.e. a complete review. That's why I believe this is not something reasonable to ask during a discussion phase.

**Strengths:**

- Relaxing the invertibility of the mixing function in identifiable representation learning is a very important direction and the suggestion that temporal context could be used to infer the latent factors in that case makes a lot of sense intuitively.
- This suggestion is novel AFAIK.
- Very few theoretical works of this nature present experiments on realistic data to show the usefulness of their approach, as was done in the present work. This is appreciated.

**Weaknesses:**

### **Is there a mathematically explicit example of model that satisfies the assumptions?**
The authors assume a standard data generating process where $z^t$ follows some dynamical process and where $x_t = g(z_t)$. However, they do not assume that the mixing $g$ is injective. Instead, they assume that there exists a map $m$ s.t. $z_t = m(x_{t:t-\mu})$. This feels like a reasonable assumption, however, I think the authors should provide at least one mathematically concrete example where this assumption holds (specifying what is f, g and m explicitly). This is important, in my opinion, to make sure that the result is not completely vacuous in the sense that there exists no model that satisfies the assumption of the theory. Right now it's not entirely clear to me that such an example exists.

### **Math mistakes, unclear proofs and bad terminology**
- Beginning of 2.1: wrong definition of surjectivity. What you describe is simply the definition of a function (i.e. it has a unique output). A map $f: A \rightarrow B$ is surjective if, for all $b \in B$, there exists $a \in A$ s.t. $f(a) = b$.
- In the third assumption of Theorem 1:
    - What is a “continuous manifold”? Can you refer to a definition from a math textbook? You mean a topological manifold? Does it mean the support of $\hat z$ can have a lower dimension than the ambient space?
    - requiring that $m, \hat m, g, \hat g$ are twice differentiable is clear, but the “i.e.” following is confusing. You could simply get rid of it.
- I’m confused by the fact that the theorems do not reuse Definition 1 with its notion of “observational equivalence”. Instead, the theorems start with $x_t = \hat g(\hat z_t)$ and $\hat z_t = \hat m (x_{t:t-\mu})$. It certainly implies observational equivalence, but is it equivalent? Should I think of the equalities here as “have equal distribution”, or is it a normal equality? Also the theorem does not refer to the data generating process of section 2.1. This is unclear.
- Definition 1: It looks like it is implicitly assumed that the random vector x_1, … x_T has a density (w.r.t. The Lebesgue measure). I think this is also assumed in the proof, equation (17), where the change of variable formula for densities is used (which works only for densities). It’s not clear that the random vector x_1, … x_T has a density. For example, if dim(x) > dim(z), it won’t be the case. Are you assuming dim(x) = dim(z)? I couldn’t find dim(x) anywhere.
- Definition 1: The authors seem to include $m$ in the parameters of the generative model. I find this a bit weird since the model is fully specified by f, p, g. No need for m in the parameters.
- Corollary 1: Usually, corollaries are very simple consequences of a theorem. Here, it doesn’t look like it’s a simple consequence, it actually looks like a generalization. Also, would it be possible to unify Theorem 1 and Corollary 1 in a way that both of these results are special cases? Suggesting because restating almost identical assumptions looks a bit inefficient.
- Equation (3), should be $\forall x_{t, t-\mu} \in \mathcal{X}^{\mu +1}$.
- Section 3.3: The terminology used here does not align with standard terminology used in topology. For example, what the authors call a “continuous domain” or a “continuous set” is usually called a “path-connected” set in topology. Please use existing terminology.
- In the Jacobian on page 7, what do the “*” mean? Zeros?
- VAE-based approach: “To uphold the conditional independence assumption, we aim to minimize the KL divergence between the posterior for each time step, p(\hat zt|xt:t−µ), and the prior distribution p(\hat zt|\hat zt−1:t−τ ).” IMO, this shows a poor understanding of what VAE’s are all about. First, for p(\hat zt|xt:t−µ), the letter “q” should be used to specify that this is not the “actual” posterior of the model, but a variational approximation. Secondly, saying the KL enforces conditional independence is weird. Conditional independence is hard-coded in your generative model, the KL is just part of your evidence lower bound. It’s not present specifically to enforce or encourage conditional independence.


### **Issues in Section 3.3**
- The authors rightfully points out that one has to be careful when going from “Jacobian is a permutation-scaling matrix” to “the mapping is a permutation composed with an element-wise transformation” when the domain of the function is not simply $\mathbb{R}^n$. However, [4] already made that point (see beginning of Section 3.1 and the discussion surrounding what they call "local" and "global" disentanglement).
- Moreover, Example 6 from [4] presents a counterexample to Lemma 1, i.e. an example of function with a path-connected domain where the Jacobian is everywhere a permutation-scaling matrix, but the function is not “disentangled”, in the sense that it cannot be written as a permutation composed with an element-wise rescaling. This implies that Lemma 1 has to be wrong.
- I also spent some time reading the proof of Lemma 1 and it is unclear. For example, what is the “n-dimensional axis except 0”? You can also find weird terminologies which makes understanding the argument impossible. This makes me even more confident that Lemma 1 is wrong.

### **Writing is unclear/imprecise**
The overall quality of writing was low. I found many sentences that were weirdly formulated. For example:
- Not sure I understand “Non-invertibility by vision persistence” from the intro. Why does the crashing car example have vision persistence? This was not explained, no?
- “Thus, we assume that there exists a maximum time lag $\mu$ and an arbitrary nonlinear function $m$...” The word “arbitrary” shouldn’t be there.
- “In this case, there is information loss in $x_t$ due to the non-invertibility of $g$.” This is imprecise. What is meant by information here? I believe what you mean is that one cannot recover $z^t$ from $x^t$ alone.
- “We say latent causal processes are identifiable if observational equivalence can lead to identifiability of the latent variables…” This phrasing is weird. They define identifiability, but use the word “identifiability” in its definition. This should be rephrased.
- “Due to the complexity of the non-invertible mixing function, the identifiable representation does not indicate the inference function is identifiable.” I don’t understand this sentence.
- “with a function m that satisfies our assumption zt = m(xt:t−µ) in existence” Weird sentence formulation.
- Figure 3 (c), what is the x-axis?

### **Should make more connections with existing works.**
- Theorem 1 seems to reuse assumptions very similar to [2], which itself reuses assumptions similar to the line of Aapo Hyvarinen’s group, see for example [3]. I think this resemblance should be highlighted in the text to help the reader understand what is truly novel in the proposed theoretical results. In general, I feel like the results could be contextualized in the literature a bit more.
- [1] should be cited, as it was among the first work showing identifiability was possible in dynamical latent dynamical systems.






[1] S. Lachapelle, P. Rodriguez Lopez, Y. Sharma, K. E. Everett, R. Le Priol, A. Lacoste, and S. Lacoste-Julien. Disentanglement via mechanism sparsity regularization: A new principle for nonlinear ICA. In First Conference on Causal Learning and Reasoning, 2022.

[2] W. Yao, G. Chen, and K. Zhang. Temporally disentangled representation learning. In Advances in Neural Information Processing Systems, 2022a.

[3] A. Hyvarinen, H. Sasaki, and R. E. Turner. Nonlinear ica using auxiliary variables and generalized contrastive learning. In AISTATS. PMLR, 2019.

[4] S. Lachapelle, D. Mahajan, I. Mitliagkas and S. Lacoste-Julien. Additive Decoders for Latent Variables Identification and Cartesian-Product Extrapolation. NeurIPS 2023.

**Questions:**

Is the advantage of the proposed algorithm over the baselines on the QA benchmark due to disentanglement and better identifiability? Or is it due to architectural choices? I feel this should be addressed, since the paper is very much centered around disentanglement and identifiability.

---

> ### Author Response · Authors · 2023-11-18
> **Response to Reviewer ur5z Part 1**
>
> Dear Reviewer ur5z, we highly appreciate all your time dedicated to reviewing this paper! Your careful reading and valuable suggestions definitely helped improve the paper's quality.  We provided the point-to-point responses below and modified our manuscript accordingly.
>
> > **W1:** Is there a mathematically explicit example of model that satisfies the assumptions? The authors assume a standard data generating process where $z_t$ follows some dynamical process and where $x_t=g(z_t)$. However, they do not assume that the mixing $g$ is injective. Instead, they assume that there exists a map $m$ s.t. $z_t=m(x_{t:t-\mu})$. This feels like a reasonable assumption, however, I think the authors should provide at least one mathematically concrete example where this assumption holds (specifying what is f, g and m explicitly). This is important, in my opinion, to make sure that the result is not completely vacuous in the sense that there exists no model that satisfies the assumption of the theory. Right now it's not entirely clear to me that such an example exists.
>
> **A1:** We sincerely appreciate your insightful suggestions. For a better understanding of when such a  condition can be satisfied, we add Section 2.3 for a detailed example. A figurative example with a fast-moving ball and a mathematical example are provided in this section.
>
> **Notice:** Due to the issue of markdown for complex equations, we use   $a\underline{ }(b)$ to represent $a_b$.
>
> Here is a mathematical example to demonstrate the existence of function $\mathbf{m}$.  Following the concept of visual persistence, let the current observation be a weakened previous observation overlaid with the current image of the object, i.e., $\mathbf{x}\underline{ }t = \mathbf{z}\underline{ }t + \frac{1}{2}\mathbf{x}\underline{ }{(t-1)} = \sum_{i=1}^{\infty} \left(\frac{1}{2}\right)^{i} \mathbf{z}\underline{ }{(t-i)}$.  Given an extra observation, the current latent variable can be rewritten as $\mathbf{z}\underline{ }t  = \mathbf{x}\underline{ }t - \frac{1}{2}\mathbf{x}\underline{ }{(t-1)}$.
> Thereby we can easily recover latent variables that cannot be obtained from a single observation, i.e., $\mathbf{z}\underline{ }t = \mathbf{m}(\mathbf{x}\underline{ }{(t:t-1)}) = \mathbf{x}\underline{ }t - \frac{1}{2}\mathbf{x}\underline{ }{(t-1)}$.
>
> > **W2:**  Beginning of 2.1: wrong definition of surjectivity. What you describe is simply the definition of a function (i.e. it has a unique output). A map $f:A\rightarrow B$ is surjective if, for all $b\in B$, there exists $a\in A$ s.t. $f(a)=b$.
>
> **A2:** We greatly appreciate your pointing out the inaccuracies in our paper. We intended to convey that the range of the mixed function $g$ is $\mathcal{X}$. However, we acknowledge that such phrasing can increase the difficulty of reading the paper. Therefore, we have revised Section 2.1 and removed the related statement.
>
> > **W3:** In the third assumption of Theorem 1:  What is a “continuous manifold”? Can you refer to a definition from a math textbook? You mean a topological manifold? Does it mean the support of $\hat{z}$
>  can have a lower dimension than the ambient space? Requiring that $m,\hat{m}, g, \hat{g}$ are twice differentiable is clear, but the “i.e.” following is confusing. You could simply get rid of it.
>
> **A3:** Thank you for your valuable comments which help clarify terminology. We used ''continuous manifold'' to represent a continuous domain. We agree that the ''path-connected domain'' is a better expression and correct it in the revised paper. We also remove the ''i.e.'' part following the sentence in Theorem 1. It makes our theorem much more concise.
>
> > **W4:** I’m confused by the fact that the theorems do not reuse Definition 1 with its notion of “observational equivalence”. Instead, the theorems start with $x_t=\hat g(\hat z_t)$ and $\hat z_t=\hat m (x_{t:t-\mu})$. It certainly implies observational equivalence, but is it equivalent? Should I think of the equalities here as “have equal distribution”, or is it a normal equality? Also the theorem does not refer to the data generating process of section 2.1. This is unclear.
>
> **A4:** Thanks for your insightful suggestions. They definitely improve the readability of our paper.
> - As you mentioned, the condition given in Theorem 1 is observational equivalent to the generating process. We have revised Theorem 1 and reused the definition of observational equivalence in the theorem.
> - The equalities in theorem 1 are normal equalities. When the normal equalities are established everywhere, the equality of distributions is also established. Thus it yields the following proof.
> - When it comes to the true generating process, we are actually talking about the process mentioned in Eq.4.

---

> ### Author Response · Authors · 2023-11-18
> **Response to Reviewer ur5z Part 2**
>
> > **W5:** Definition 1: It looks like it is implicitly assumed that the random vector x\underline{ }1, … x\underline{ }T has a density (w.r.t. The Lebesgue measure). I think this is also assumed in the proof, equation (17), where the change of variable formula for densities is used (which works only for densities). It’s not clear that the random vector x\underline{ }1, … x\underline{ }T has a density. For example, if dim(x) > dim(z), it won’t be the case. Are you assuming dim(x) = dim(z)? I couldn’t find dim(x) anywhere.
>
> **A5:** Thanks for your valuable question. The observation $x_t$ is indeed a random vector with a density, and it is also assumed in Eq.18 (originally Eq.17). In my opinion, whether $dim(x)=dim(z)$ doesn't matter, since we never assume there is an equation like $p(z)/|J_g|=p(x)$ when $x=g(z)$. Thus, we do not require jacobian $J_g$ of mixing function $g$ to be a square matrix. That is to say, $dim(x)$ can be different from $dim(z)$.
>
> > **W6:** Definition 1: The authors seem to include $m$ in the parameters of the generative model. I find this a bit weird since the model is fully specified by f, p, g. No need for m in the parameters.
>
> **A6:** We greatly appreciate your suggestions. Although $\mathbf{m}$ does not explicitly participate in the generation process, it does so implicitly as a property of the mixing function $\mathbf{g}$. That is why we put $\mathbf{m}$ into the definition part in the manuscript. However, we do agree that this may cause confusion. Therefore, we have removed $\mathbf{m}$ from the definition part and added an explanation in the paragraph below Definition 1.
>
> > **W7:** Corollary 1: Usually, corollaries are very simple consequences of a theorem. Here, it doesn’t look like it’s a simple consequence, it actually looks like a generalization. Also, would it be possible to unify Theorem 1 and Corollary 1 in a way that both of these results are special cases? Suggesting because restating almost identical assumptions looks a bit inefficient.
>
> **A7:** Thank you for your insightful suggestions. We used to split them up because they can fit into the two scenarios named "UG" and "UG-TDMP" separately. We totally agree that a merged version is more clear. In light of your suggestion, we have merged them into Theorem 1 in Section 3.1 in the revised paper.
>
> > **W8:** Equation (3), should be $\mathbf{x}_{t:t-\mu} \in \mathcal{X}^{\mu+1}$.
>
> **A8:** Thanks for your correction. We have corrected it in Eq.3.
>
> > **W9:** Section 3.3: The terminology used here does not align with standard terminology used in topology. For example, what the authors call a “continuous domain” or a “continuous set” is usually called a “path-connected” set in topology. Please use existing terminology.
>
> **A9:** Thank you very much for your feedback on the readability of our article. The reason we use “continuous set” is that we start to consider this issue from a perspective of the property of functions, rather than a perspective of topology. We do agree that a more accurate terminology can better improve our paper. Thus we revise the manuscript accordingly, changing all "continuous set/domain" to "path-connected" in all theorems, lemmas, explanations, and proof. Thanks for your effort to enhance the readability of our paper again.
>
> > **W10:**  In the Jacobian on page 7, what do the “*” mean? Zeros?
>
> **A10:** Thank you for your pointing out this typo. We have corrected it to zeros.
>
> > **W11:** VAE-based approach: “To uphold the conditional independence assumption, we aim to minimize the KL divergence between the posterior for each time step, p(\hat zt|xt:t−µ), and the prior distribution p(\hat zt|\hat zt−1:t−τ ).” IMO, this shows a poor understanding of what VAE’s are all about. First, for p(\hat zt|xt:t−µ), the letter “q” should be used to specify that this is not the “actual” posterior of the model, but a variational approximation. Secondly, saying the KL enforces conditional independence is weird. Conditional independence is hard-coded in your generative model, the KL is just part of your evidence lower bound. It’s not present specifically to enforce or encourage conditional independence.
>
> **A11:** We respectfully appreciate your insightful suggestions. For the variational approximation, we correct the wrong symbol as $q(\hat z_t|x_{t:t−µ})$ in Section 4. Besides, it is totally right that the conditional independence is hard-coded in our generative model by $\hat \epsilon_{it} = f_i^{-1}(\hat z_{it},\hat z_{t-1:t-\tau})$.
>
> In fact,   we propose to minimize the KL divergence between the posterior distribution  $q(\hat z_t|x_{t:t-\mu})$, and a prior distribution $p(\hat z_t|\hat z_{t-1:t-\tau})$, for out conditional independence assumption. By hard-coding the prior distribution to be conditionally independent on $\hat z_{t-1:t-\tau}$, we expect the posterior to also be subject to the conditional independence assumption.

---

> ### Author Response · Authors · 2023-11-18
> **Response to Reviewer ur5z Part 3**
>
> > **W12:** The authors rightfully points out that one has to be careful when going from “Jacobian is a permutation-scaling matrix” to “the mapping is a permutation composed with an element-wise transformation” when the domain of the function is not simply $\mathbb{R}^n$. However, [4] already made that point (see beginning of Section 3.1 and the discussion surrounding what they call "local" and "global" disentanglement).
>
> **A12:**  We highly appreciate the sharing of this paper. We had learned a lot from this. Besides, we want to express our sincere gratitude for your recognition of our contribution to this. Please kindly note that the publication date of NeurIPS23 and the submission date of ICLR24 are very close, leading us to miss this good reference. To make the contribution more clear, we have added the citation, included the discussion, and suggested the readers refer to this paper for details. Please kindly refer to the discussion under Lemma 1 in Section 3.2 in the revised version.
>
> > **W13:** Moreover, Example 6 from [4] presents a counterexample to Lemma 1, i.e. an example of function with a path-connected domain where the Jacobian is everywhere a permutation-scaling matrix, but the function is not “disentangled”, in the sense that it cannot be written as a permutation composed with an element-wise rescaling. This implies that Lemma 1 has to be wrong.
>
> **A13:** After carefully looking into example 6 of [4], we respectfully believe that this example is not a counterexample to Lemma 1. It is because we assumed that the domain of $h$ is path-connected. In this example, $\mathcal{Z}=\mathcal{Z}^{(1)} \cup \mathcal{Z}^{(2)} \subseteq \mathbb{R}^2$ where $\mathcal{Z}^{(1)} = ${ $\mathbf{z} \in  \mathbb{R}^2 | z_2 \leq 0 $ } and $\mathcal{Z}^{(2)} = ${ $\mathbf{z} \in  \mathbb{R}^2 | z_2 \geq 1 $ }, is the disconnected domain.  Thus, it is not a counterexample to Lemma 1.
>
> > **W14:** I also spent some time reading the proof of Lemma 1 and it is unclear. For example, what is the “n-dimensional axis except 0”? You can also find weird terminologies which makes understanding the argument impossible. This makes me even more confident that Lemma 1 is wrong.
>
> **A14:** Many thanks for your effort to help us to improve readability. We replace the inaccurate terminologies and reorganize the proof to make it more clear. Sorry for the inconvenience caused by the unclear proof. And we do value your effort to enhance the readability of our paper.  Please refer to the proof of Lemma 1 in the revised manuscript on page 18, and kindly let us know if there is anything not clear in our proof.

---

> ### Author Response · Authors · 2023-11-18
> **Response to Reviewer ur5z Part 4**
>
> > **W15:** Not sure I understand “Non-invertibility by vision persistence” from the intro. Why does the crashing car example have vision persistence? This was not explained, no?
>
> **A15:**  Thanks for your insightful question. Actually, all the objects are able to have the vision persistence. But, in the context of high-speed movement, this phenomenon is more obvious. Here, we use this example since there is a motion blur (one special example of vision persistence) caused by high speed movement of crashing car. We have included these explanations in the caption of Figure 1 revised paper.
>
> > **W16:** “Thus, we assume that there exists a maximum time lag $\mu$ and an arbitrary nonlinear function m...” The word “arbitrary” shouldn’t be there.
>
> **A16:** Many thanks for your helpful feedback. We remove “arbitrary” in the paragraph above Eq.2 in Section 2.1 to make it more clear.
>
> > **W17:** “In this case, there is information loss in $x\underline{ }t$ due to the non-invertibility of g.” This is imprecise. What is meant by information here? I believe what you mean is that one cannot recover $z\underline{ }t$ from $x\underline{ }t$ alone.
>
> **A17:** Many thanks for your helpful feedback. We replace the original sentence with "In this case, one cannot recover $\mathbf{z}_t$ from $\mathbf{x}_t$ alone due to the non-invertibility of $\mathbf{g}$." in the paragraph above Eq.2 in Section 2.1 to improve the readability.
>
> > **W18:** “We say latent causal processes are identifiable if observational equivalence can lead to identifiability of the latent variables…” This phrasing is weird. They define identifiability, but use the word “identifiability” in its definition. This should be rephrased.
>
> **A18:** Thank you for pointing out the issue of circular definition in our work. In light of your suggestion, we rephrase it as "We say latent causal processes are identifiable if observational equivalence can lead to a version of latent variable $z_t = m(x_{t:t-\mu})$ up to permutation $\pi$ and component-wise invertible transformation $T$."
>
> > **W19:** “Due to the complexity of the non-invertible mixing function, the identifiable representation does not indicate the inference function is identifiable.” I don’t understand this sentence.
>
>  **A19:** Thank you for the feedback. Originally we want to say that since function $g,m$ are not invertible themselves, the identifiability of $g,m$ can not be guaranteed. Meanwhile, the latent variable $z_t$ is identifiable. Thank you for the effort to improve the quality of our writing. We revised the paragraphs at the end of Section 2.2 for better readability.
>
> > **W20:** “with a function m that satisfies our assumption zt = m(xt:t−µ) in existence” Weird sentence formulation.
>
>  **A20:** Thanks for the suggestion. We revise the sentence as "where a function $\mathbf{m}$ satisfying $z_t = m(x_{t:t-\mu})$ exists." after Eq.4 in section 3.1.
>
> > **W21:** Figure 3 (c), what is the x-axis?
>
> **A21:** The x-axis in Figure 4 (original Figure 3) is the epochs$/(10^{-1})$
>
> > **W22:** Theorem 1 seems to reuse assumptions very similar to [2], which itself reuses assumptions similar to the line of Aapo Hyvarinen’s group, see for example [3]. I think this resemblance should be highlighted in the text to help the reader understand what is truly novel in the proposed theoretical results. In general, I feel like the results could be contextualized in the literature a bit more.
>
>  **A22:** Thank you for pointing out that it's important to emphasize the relationship between these works and highlight their inheritances. We highlight it in Section 3.1, after where Theorem 1 is introduced.
>
> > **W23:** [1] should be cited, as it was among the first work showing identifiability was possible in dynamical latent dynamical systems.
>
> **A23:** Thanks very much for recommending this paper. It is helpful to make the background clearer. We have included this in the introduction of the revised version.
>
> > **W24(Q1):** Is the advantage of the proposed algorithm over the baselines on the QA benchmark due to disentanglement and better identifiability? Or is it due to architectural choices? I feel this should be addressed, since the paper is very much centered around disentanglement and identifiability.
>
> **A24:** Thanks for this insightful question. In real-world applications, we have no ground truth for computing metrics like MCC. To address this, we use 2 proxy metrics including the disentanglement and the reconstruction ability of the learned representations. In practice, we employ ELBO loss as the metric. We provide results in Appendix A4.2 and Figure A2. The results show that our methods exhibit superior performance in both disentanglement and reconstruction ability compared to baseline methods. It can serve as side evidence to support the advantage of better identifiability.

---

> ### Author Response · Authors · 2023-11-21
> **Could you please let us know whether our responses and updated submission properly addressed your concern?**
>
> Dear Reviewer ur5z,
>
> We express our sincere gratitude for taking the time to review our manuscript. Your suggestions regarding the theoretical aspects and readability of our article have greatly contributed to improving its quality. We have made detailed revisions to the manuscript and addressed your questions in our response. We hope that our answers have addressed any concerns you had regarding our work. Your feedback is vital to us, and any response would be further appreciated.
>
> Many thanks,
>
> Authors of submission 2088

---

> > ### Comment · Reviewer_ur5z · 2023-11-21
> >
> > Thanks for carefully considering my concerns. I apologize for the delayed answered, I thought we would have more discussion time, like last year.
> >
> > Regarding the explicit model satisfying the assumptions: This example is not transparent to me. What are the functions $g$, $f_i$ and $p_{\epsilon_i}$ as introduced in (1) here? What is the corresponding $m$?
> >
> > Density of x_t: Equation (15) in the new manuscript assumes a density w.r.t. z_t, x_t-1, .... . This is a bit unclear.
> >
> > "Please kindly refer to the discussion under Lemma 1 in Section 3.2 in the revised version.": I'm didn't find the discussion under Lemma 1, I think you meant above.
> >
> > "After carefully looking into example 6 of [4], ..." Oops, it looks like this paper has been updated on arxiv since I wrote my review such that the examples numbering changed. I was referring to Example 6 from this version: https://arxiv.org/abs/2307.02598v1. (in the recent version it's Example 7).
> >
> > Thanks again for considering my concerns seriously. I won't change my score, but I want to reiterate that I very much love the direction proposed in this work. The assumption that the current latent can be inferred when using past observations makes a lot of sense to me. My issues have to do with the execution of the idea + technical details that are not properly treated. I sincerely believe that with some efforts, this work can get well above the acceptance threshold.

---

> > > ### Author Response · Authors · 2023-11-22
> > > **Further response to reviewer ur5z**
> > >
> > > > **Q1:** Regarding the explicit model satisfying the assumptions: This example is not transparent to me. What are the functions $g$, $f_i$, and $p_{\epsilon_i}$ as introduced in (1) here? What is the corresponding $m$?
> > >
> > > **A1:** Thanks for your question.
> > > - For $g$, we have $x_t = g(z_{<t}) = z_t + \frac{1}{2}x_{t-1} = \sum_{i=1}^{\infty} \left(\frac{1}{2}\right)^{i} z_{t-i}$, as introduced from a simplified visual persistance scenario;
> > > - for $m$, we have $z_t = m(x_{t:t-1}) = x_t - \frac{1}{2}x_{t-1}$;
> > > - for $f_i$ and $p_{\epsilon_i}$, they are not necessarily to be specified in this example, as long as the sufficiency assumption can be satisfied. It is ideal since the ball can undergo any accelerated/decelerated linear/curved motion. As a most simplified example, consider $f_i$ as $z_t = \sum_{i=1}^{\infty} \left(\frac{1}{2}\right)^{i} z_{t-i} + \epsilon_{it}$, where $\epsilon_{it}$ to be an unique Laplace distribution.
> > >
> > > > **Q2** Density of x_t: Equation (15) in the new manuscript assumes a density w.r.t. z_t, x_t-1, .... . This is a bit unclear.
> > >
> > > **A2:** Thanks for your valuable question. By Equation (15), we do assume that a density with regard to both $z_t$ and $x_t$. Although we use normal equality in definition (1) while using density in Equation (15), indeed the density equations in Equation (15) is a corollary of the normal equalities in Definition (1).
> > >
> > > - By normal equality in Definition (1), we would like to show the causal relationship between variables. For each sample of the process $\{z_{1:T},x_{1:T}\}$ in Definition (1), any sampled varibles $z_t,x_t$ will follow equations in Definition (1).
> > >
> > > - By density equality in Equation (15), we would like to show the change of joint distribution of several causally related variables. As we have answered in **A4**, "the equalities in theorem 1 are normal equalities. When the normal equalities are established everywhere, the equality of distributions is also established. Thus it yields the following proof." To clarify, the Jacobian in Equation (15) is related to the following normal equality
> > >
> > > $$
> > > \begin{bmatrix}
> > > \hat z_t \newline
> > > x_{t-1:t-\mu-r-1}
> > > \end{bmatrix} =
> > > \begin{bmatrix}
> > > \hat m(g(z_t,m(x_{t-1:t-\mu-1}), \cdots, m(x_{t-r:t-\mu-r})), x_{t-1:t-\mu})
> > > \newline x_{t-1:t-\mu-r-1}
> > > \end{bmatrix}
> > > $$
> > >
> > > For one more example, consider a causal process $Y=X$. Here we have a normal equation universally established for any sample pair of $(x,y)$, i.e., $x=y$. This can immediately yield the equality on distribution, i.e., $P(Y)=P(X)/|I|$.
> > >
> > > I hope this will help to solve your concerns. If there is anything still remaining not clear, please do not hesitate to let us know.
> > >
> > > > **Q3:** "Please kindly refer to the discussion under Lemma 1 in Section 3.2 in the revised version.": I'm didn't find the discussion under Lemma 1, I think you meant above.
> > >
> > > **A3:** We apologize for the poor reading experience caused, as we have made further adjustments to the formatting of the paper and have not had a chance to rewrite the response. You are correct, we placed it above Lemma 1 in the latest version.
> > >
> > > > **Q4:** "After carefully looking into example 6 of [4], ..." Oops, it looks like this paper has been updated on arxiv since I wrote my review such that the examples numbering changed. I was referring to Example 6 from this version: https://arxiv.org/abs/2307.02598v1. (in the recent version it's Example 7).
> > >
> > > **A4:** Thank you for the correction for the index of the example. Let us consider Example 7 in this paper, i.e.,
> > > $$
> > > f(z):= \begin{bmatrix}z_1 \newline z_1^2 \newline z_1^3 \newline z_1^4 \newline \end{bmatrix} +
> > > \begin{bmatrix}(z_2+1) \newline (z_2+1)^2 \newline (z_2+1)^3 \newline (z_2+1)^4 \newline \end{bmatrix}.
> > > $$
> > >
> > > We would like to point out that Example 7 is not in the scope of our lemmas.
> > > - The function in Example 7 is $f:\mathbb{R}^2\rightarrow\mathbb{R}^4$, while we require that the input and output dimensions should be the same in Lemma 1. Thus, the jacobian must be a square matrix.
> > > - Even though the jacobian is a square matrix, we also require that there exist at most one non-zero entry in each row and column. It is also not satisfied with this assumption.
> > > In conclusion, it is not a counterexample for Lemma 1.
> > >
> > > **Finally**, we are more than thankful for your suggestions to improve the quality of our paper. As mentioned by reviewers WQAA, 5cYn, and uMXb, we made significant improvements to our revised version. We believe that the current version and the direction we are discussing deserve to be noticed. Meanwhile, thanks for your new questions again. Please let us know if you have any further concerns. We hope to have the opportunity to address all your concerns.

---

> > > > ### Comment · Reviewer_ur5z · 2023-11-22
> > > >
> > > > Q1: I think you have to specify f_i and epsilon in your example, otherwise it's incomplete. I'm still left unsure whether the theory is vacuous or not (to be fair, I don't think it is vacuous, but I think it's important to prove it)
> > > >
> > > > Q4: Copy-pasted from my previous message: "I was referring to Example 6 from this version: https://arxiv.org/abs/2307.02598v1". Looks like you are talking about the wrong example here.
> > > >
> > > > I'm happy to see that a very large fraction of the paper has been rewritten to integrate the comments of the reviewers. However, I believe this is much more revisions that is reasonable to review during a discussion phase like this one.

---

> > > > > ### Author Response · Authors · 2023-11-23
> > > > > **Further response to reviewer ur5z (2)**
> > > > >
> > > > > > **A1:** Copy-pasted from my previous response: "As a most simplified example, consider $f_i$ as $z_t = \sum_{i=1}^{\infty} \left(\frac{1}{2}\right)^{i} z_{t-i} + \epsilon_{it}$, where $\epsilon_{it}$ to be an unique Laplace distribution." If it still not clear to you, could you please further explain why it is still incomplete.
> > > > >
> > > > > > **A4:** Sorry for the confusion. We mistakely took the wrong example. With regard to example 6 in this paper as
> > > > > $$
> > > > > v_1(z) := z_1, \quad v_2(z) :=
> > > > > \begin{cases}
> > > > > \frac{(z_2+1)^2+1}{2} & if z\in \hat Z^{(b)} \newline
> > > > > \frac{e^{z_2}}{2} & if z\in \hat Z^{(o)}
> > > > > \end{cases},
> > > > > $$
> > > > >
> > > > > we would like to argue that we have a different definition on "global disentangment". In the setting of our Lemma 1, we only prove that the permutaion is fixed. However in this example, the permutation is fixed, while the nonlinear element-wise function will change. This issue happens because the gap between "global disentanglement" in this paper and our definition on disentanglement. We are refining it in our paper to make it more clear.
> > > > >
> > > > > In our Lemma, we provide assumptions under which the permutation can be fixed, while allowing the component-wise function changes. We focus more on permutaion, because once it fixed, the "information" will be assigned to a "position" such that a disentangled edition and perception can be operated given any example.
> > > > >
> > > > > Thanks for your valuable example again.

---

> > > > > ### Author Response · Authors · 2023-11-23
> > > > > **What do you think of giving your recommendation based on the latest resubmission?**
> > > > >
> > > > > Thank you very much again for your constructive feedback during the discussion. As you see, we are lucky to have the opportunity to improve the quality of the work efficiently and effectively.  As researchers, we appreciate your engagement and encouragement.
> > > > >
> > > > > Since we already made major changes to the submission, it would be highly appreciated if you could consider the current version of the submission and provide the final recommendation based on it, for a smoother dissemination  and communication of research findings.  If you have further feedback, we will be eager to see it.  Thanks!
> > > > >
> > > > >
> > > > > With best wishes,
> > > > >
> > > > > Authors of #2088

---

### Official Review · Reviewer_5cYn · 2023-10-31

**Soundness:** 3 good
**Presentation:** 3 good
**Contribution:** 2 fair
**Rating:** 5
**Confidence:** 3

**Summary:**

The work studies an identifiability theory of recovering causal latent variables in a non-invertible generation process. The theoretical results show the causal latent variable is identifiable up to permutation and a component-wise transformation under certain conditions. Based on the theoretical study results, the work proposes, CaRiNG, which extends Sequential VAE with a normalizing flow in the latent transition dynamics and an encoder incorporating history information. CaRiNG demonstrates superior performance to baseline methods on synthetic  tasks aligning with the theoretical study. In a real-world experiment setting of understanding traffic dynamics, the proposed approach also demonstrates competitive performance against baseline approaches.

**Strengths:**

The propose approach is backed by solid theoretical study on the identifiability of latent causal variables; the theoretical study results are also well supported by experiment results under carefully designed synthetic settings.

**Weaknesses:**

1. The proposed approach, CaRiNG, is not significantly different from the original Sequential VAE[1], especially from a probabilistic model perspective. There are also existing VAE works[2, 3] that incorporate normalizing flows. The novelty of CaRiNG as a new approach is rather limited.
2. The presentation of the work needs improvements. The lack of explicit connections between theoretical study (Sec. 3) and model design (Sec. 4) makes the work less readable. In other words, I would suggest the reader to directly connects their model design choices in Sec. 4 to the conditions of their theoretical results in Sec. 3. Moreover, Sec. 3.3 and Sec. 3.4 are primarily supporting or supplementing the theoretical results in Sec. 3.1 and Sec. 3.2 but not critical to the identifiability theory's presentation or the proposal of CaRiNG. Their positioning in the work is distracting in my personal opinion and much of the detailed discussions in Sec. 3.3 and Sec. 3.4 can be moved to the appendix.
3. The work repeatedly claims the guarantee of identifiability or guarantee of identifiability under mild conditions. However, their theoretical results also rely on the existing of a function $m$ such that $z_t = m(x_{t:t-\mu})$. It is not clear if this existence condition can be trivially satisfied, especially in real-world settings, including the work's real-world experiment. Even if such a function exists, it is also not clear how to determine $\mu$.
4. The work studies the proposed approach on only one real-world dataset and relies on QA accuracy as a proxy to indirectly evaluate the model's ability to understand the underlying causality. Even though it is challenging to evaluate the identifiability of causal latent dynamics, experiments on different real-world data and different proxy metrics could provide more convincing results.

[1] Chung, Junyoung, et al. "A recurrent latent variable model for sequential data." Advances in neural information processing systems 28 (2015).

[2] Rezende, Danilo, and Shakir Mohamed. "Variational inference with normalizing flows." International conference on machine learning. PMLR, 2015.

[3] Ziegler, Zachary, and Alexander Rush. "Latent normalizing flows for discrete sequences." International Conference on Machine Learning. PMLR, 2019.

**Questions:**

Apart from the points in *Weaknesses*, I also have the following questions and suggestions:
1. Sequential VAE can be viewed as a degenerate version of CaRiNG where prior distribution is another Gaussian with non-zero mean and diagonal variance. It is a valid baseline to compare against and the comparison could also help the work better demonstrate the importance of the proposed design changes of CaRiNG from Sequential VAE.
2. The transition lag $\tau$ is an important hyper-parameter of the proposed approach. The work includes ablation study results on the choice of $\tau$ in controlled synthetic setting. It is actually more important to do hyper-parameter search over its values in real-world settings where we do not know the true underlying generative process to avoid model mis-specification.

---

> ### Author Response · Authors · 2023-11-18
> **Response to Reviewer 5cYn Part 1**
>
> Dear Reviewer 5cYn, we are grateful for your valuable suggestions which helped improve the quality of our paper. In light of your suggestions, we revised our main paper and appendix to better highlight the contribution and explain the details. Below, we provide point-to-point responses.
>
> > **W1:** The proposed approach, CaRiNG, is not significantly different from the original Sequential VAE[1], especially from a probabilistic model perspective. There are also existing VAE works[2, 3] that incorporate normalizing flows. The novelty of CaRiNG as a new approach is rather limited.
>
> **A1:** Thanks for your comments which helped highlight our unique contribution.
>
> **Notice:** Due to the issue of markdown for complex equations, we use   $a\underline{ }(b)$ to represent $a_b$.
>
> **Technical differences**
> Compared with the original Sequential VAE, we would like to highlight the following differences in our approach:
> 1) We established the identifiability theory under the non-invertibility mixing functions, which complements the existing body of the ICA framework.
> 2) We model the temporal dynamics of multivariate with a transition function.
> 3) We added a prior module to encourage the 'conditional independence' of the learned latent variables.
> Please kindly note that, our goal is to learn the causal representations with identifiability guarantees, which is different from the conventional VAE-based methods towards better generation ability.
> Our contributions lie in investigating what assumptions we can achieve this identifiability and how to do it in practice.
>
> **Experimental comparison**
> Experimentally, we also compared the identifiability of the learned variables between our methods and one of the advanced SequentialVAE methods, SKD[R1], using the synthetic data.  In light of your advice, we further add the comparisons with the original SequentialVAE method. As shown in Table 1 in the revised paper, our method shows significant improvement over SKD. Interestingly, we find that SequentialVAE method works better than other methods which don't use the temporal context, which also demonstrates the necessity of temporal context to solve the invertibility issue. However, we still find that constraining conditional independence benefits better performance.
>
> **Comparison with other VAE works[2, 3] that incorporate normalizing flows**  Though both our method and some existing works[2, 3] applied the normalizing flow, we highlight the motivations and other model details are different.
> 1) We apply normalizing flows to help calculate the prior distribution and further constrain the conditional independence, which is not claimed as our unique contribution. Please refer to Section 4 in the revised version for details.  For existing works, they apply the normalizing flow to obtain the invertible deterministic transformation between two variables in the sequence.
> 2) We applied a sequence-to-step encoder, which leverages the temporal context to recover the information lost caused by non-invertibility mixing functions. The existing works[2, 3] only focus on the current observations.
>
> We have added these discussions in the revised Appendix A6.1 and A6.2. Please kindly let us if you have any further questions.
>
> [R1] Berman N, Naiman I, Azencot O. Multifactor sequential disentanglement via structured koopman autoencoders[J]. arXiv preprint arXiv:2303.17264, 2023.

---

> ### Author Response · Authors · 2023-11-18
> **Response to Reviewer 5cYn Part 2**
>
> > **W2:** The presentation of the work needs improvements. The lack of explicit connections between theoretical study (Sec. 3) and model design (Sec. 4) makes the work less readable. In other words, I would suggest the reader to directly connects their model design choices in Sec. 4 to the conditions of their theoretical results in Sec. 3. Moreover, Sec. 3.3 and Sec. 3.4 are primarily supporting or supplementing the theoretical results in Sec. 3.1 and Sec. 3.2 but not critical to the identifiability theory's presentation or the proposal of CaRiNG. Their positioning in the work is distracting in my personal opinion and much of the detailed discussions in Sec. 3.3 and Sec. 3.4 can be moved to the appendix.
>
> **A2:** We highly appreciate your efforts to improve the writing quality of this paper.  In light of your suggestions, we have modified our manuscript and appendix. We believe this version is much more readable than before.
>
> We reorganized Section 4 and explicitly connected network components and the assumptions in our theorem.
> Besides, we have moved the original Sec. 3.4 into appendix. Instead, we provide some examples to elaborate our problem setting-ups in Section 2.3, which may help with the following questions.
> For original Sec. 3.3 (now it is Sec. 3.2), we respectfully believe it fills the gaps in existing ICA literature and serves as an important lemma for proving our Theorem 1, which needs to be highlighted in the main submission. We also understand that these sections may not contribute significantly to understanding the algorithm details. Therefore, we have made efforts to streamline the content in the main body and moved some of the content to the appendix.
>
> > **W3:** The work repeatedly claims the guarantee of identifiability or guarantee of identifiability under mild conditions. However, their theoretical results also rely on the existing of a function $m$ such that $z\underline{ }t=m(x\underline{ }{(t:t-\mu)})$. It is not clear if this existence condition can be trivially satisfied, especially in real-world settings, including the work's real-world experiment. Even if such a function exists, it is also not clear how to determine $\mu$.
>
> **A3:** Thank you for your insightful comments. Please kindly note that compared to models that do not consider non-invertibility, our condition is relatively lenient.
> It shows our unique contributions to complement the existing body of Nonlinear ICA with non-invertibility. Below, we provide some intuitive and mathematical illustrations to explain why this assumption makes sense and whether this existence condition can be trivially satisfied.
>
>
> **Further illustration with intuitive examples**
> Consider a rapidly moving ball on a two-dimensional plane as described in Figure 2. The horizontal and vertical coordinates of the ball's position at any given moment can be represented by the latent variable $\mathbf{z}\underline{ }t\in \mathbb{R}^2$. We assume that the ball follows a curved trajectory constrained by the nonlinear function $\mathbf{f}$  as it moves.
>
> Suppose that we observe the ball with a visual persistence effect, where each observation $\mathbf{x}\underline{ }t$ captures several consecutive latent variables as
> $\mathbf{x}\underline{ }t = \mathbf{g}(\mathbf{z}_{<t})$.
> The mixing function $\mathbf{g}$ refers to the weighted sum of the images obtained through multiple exposures, which is what a person ultimately observes as $\mathbf{x}\underline{ }t$. In this case, the invertibility of the mapping from $\mathbf{z}\underline{ }{t}$ to $\mathbf{x}\underline{ }{t}$ is compromised since the current frame also contains the latent information from previous frames.
>
> **Further illustration with mathematical examples**
>  Following the concept of visual persistence, let the observation be  $\mathbf{x}\underline{ }t = \mathbf{z}\underline{ }t + \frac{1}{2}\mathbf{x}\underline{ }{(t-1)} = \sum_{i=1}^{\infty} \left(\frac{1}{2}\right)^{i} \mathbf{z}\underline{ }{(t-i)}$.  Given an extra observation, the current latent variable can be rewritten as $\mathbf{z}\underline{ }t  = \mathbf{x}\underline{ }t - \frac{1}{2}\mathbf{x}\underline{ }{(t-1)}$.
> Thereby we can easily recover latent variables that cannot be obtained from a single observation, i.e., $\mathbf{z}\underline{ }t = \mathbf{m}(\mathbf{x}\underline{ }{(t:t-1)}) = \mathbf{x}\underline{ }t - \frac{1}{2}\mathbf{x}\underline{ }{(t-1)}$.
>
> We have added Section 2.3 in the revised version for the above illustrations.
>
> **How to determine $\mu$**
> In practical applications, we can apply the model selection trick to choose an appropriate $\mu$ in the model implementation. In experiments, we found that the $\mu$ is not necessarily better when it's larger. As observation length increases, information recovery becomes less efficient, adding noise and impeding model training, though a larger $\mu$ implies more context information recovery.
>
> We give a simple example in Appendix A4.4.

---

> ### Author Response · Authors · 2023-11-18
> **Response to Reviewer 5cYn Part 3**
>
> > **W4:** The work studies the proposed approach on only one real-world dataset and relies on QA accuracy as a proxy to indirectly evaluate the model's ability to understand the underlying causality. Even though it is challenging to evaluate the identifiability of causal latent dynamics, experiments on different real-world data and different proxy metrics could provide more convincing results.
>
>
> **A4:** Thank you very much for your insightful suggestions. In fact, one of the main motivations of our work is to bridge the gap between assumptions made in existing methods and the challenges present in real-world scenarios. We are committed to extending our methodology to real datasets and validating it on a broader range of real-world data. Currently, we are conducting experiments on the Volleyball dataset[R2]. However, due to time constraints and the required tasks such as code migration, we have not yet finished these experiments. We are making every effort to complete this experiment before the deadline.
>
> > **W5(Q1):** Sequential VAE can be viewed as a degenerate version of CaRiNG where prior distribution is another Gaussian with non-zero mean and diagonal variance. It is a valid baseline to compare against and the comparison could also help the work better demonstrate the importance of the proposed design changes of CaRiNG from Sequential VAE.
>
>
> **A5:** We are grateful for your valuable suggestions which made our comparison more complete.  Please refer to **Answer1 Experimental comparison** part for the response. We also include this experiment in Table 1 in the revised version.
>
> > **W6(Q2):** The transition lag $\tau$ is an important hyper-parameter of the proposed approach. The work includes ablation study results on the choice of
>  in controlled synthetic setting. It is actually more important to do hyper-parameter search over its values in real-world settings where we do not know the true underlying generative process to avoid model mis-specification.
>
>  **A6:** In light of your suggestion, we provide the results of the ablation study of the real-world experiment (SUTD-TrafficQA dataset) on time lag $\tau$ here. It's clear that the results are robust to the time lag $\tau$ because all the ablation results share similar accuracies. We have included this result in Appendix A4.3 and Table A4.
>
> | $\tau$   | 1     | 2     | 3     |
> | -------- | ----- | ----- | ----- |
> | Accuracy | 41.22 | 41.23 | 41.27 |
>
> [R2] Ibrahim M S, Muralidharan S, Deng Z, et al. A hierarchical deep temporal model for group activity recognition[C]//Proceedings of the IEEE conference on computer vision and pattern recognition. 2016: 1971-1980.

---

> ### Author Response · Authors · 2023-11-21
> **Could you please let us know whether our responses and updated submission properly addressed your concern?**
>
> Dear Reviewer 5cYn,
>
> Thank you for your valuable time dedicated to reviewing our submission and for your insightful suggestions. We've tried our best to address your concerns in the response and updated submission.
> Due to the limited rebuttal discussion period, we eagerly await any feedback you may have regarding these changes. If you have further comments, please kindly let us know--we hope for the opportunity to respond to them.
>
> Many thanks,
>
> Authors of submission 2088

---

> > ### Comment · Reviewer_5cYn · 2023-11-21
> > **Post-rebuttal Response**
> >
> > I would like to thank the authors for their hard work and detailed response. The presentation of the work is much better after the significant revision. The distinction and unique contribution of the work is also much more clear from existing works. However, there are still two points I'm not satisfied with. The intuitive and mathematical examples doesn't tell us that if the assumption of in the invertibility theoretical results can be trivially satisfied [w(4)]. Moreover, the ablation study on the time lag $\tau$ doesn't provide much insights as the results are similar. [w(7)]. It could be due to the assumptions are not satisfied or the evaluation metrics are not good enough and needs further explanation.. I appreciate the authors commitment to future study on more real-world datasets. Overall I'm satisfied with the response. With argument on both sides, I think this is a borderline paper with more room of improvements.

---

> > > ### Author Response · Authors · 2023-11-22
> > > **Further response to reviewer 5cYn**
> > >
> > > > **Q1:** The intuitive and mathematical examples doesn't tell us that if the assumption of in the invertibility theoretical results can be trivially satisfied [w(4)]
> > >
> > > **A1:** Thanks for your valuable question to improve the readability.
> > > - (conditional independence) Intuitively, The ball may experience random airflow from the x and y directions while in motion, resulting in independent noise;
> > > - (sufficiency) Most of the time, the distribution of noise will not be identical to each other in the real world;
> > > - (continuity) In most real-world scenarios, the continuity is assumed to be true, which is also the foundation for most ML tasks and optimization techniques.
> > >
> > > Please note that all our assumptions, including the data generating process, follow traditional temporal nonlinear ICA framework, such as TDRL, except the generation function $g$ and its ‘inverse' $m$. By comparison, TDRL strictly requires $g$ to be invertible, and there exists an inverse function $g^{-1}$, while our method further allows a possible local non-invertibility. It implies that our assumption is much easier to be satisfied in the real world than the previous temporal nonlinear ICA framework. When $\mu=r=0$, Caring can be reduced to TDRL with the same assumptions.
> > >
> > > > Moreover, the ablation study on the time lag $\tau$ doesn't provide much insights as the results are similar [w(7)]. It could be due to the assumptions are not satisfied or the evaluation metrics are not good enough and needs further explanation.
> > >
> > > Thanks for your insightful feedback, since we did not find the w(7), we believe that you mean w(6). Similarily, we believe that the question above is about w(3).
> > >
> > > In real-world scenarios, $\tau$ is not the determinant factor for identifiability, compared to $\mu$.
> > > - $\tau$ decides how many steps will be concluded in this transition process, a false $\tau$ will consider the unmatched part as a noise. For example, in the trafficQA dataset, a car undergoing linear motion can be easily described by a transition function with the preceding one or two frames. A similar trend in the results can be found in TDRL as well in our experiments.
> > > - $\mu$ decides how many steps will be needed to recover the missing information, which is much more vital to identifiability under non-invertible scenarios. For example, if a car is occluded for 1 frame, then we need at least another 2 frames to recover the occluded car. That is why $\mu$ is vital to our identifiability.
> > >
> > > In conclusion, similar results of different $\tau$, and varied results of different $\mu$ are more consistent with our intuition. We have provided synthetic ablation in Appendix A4.4 to discuss it.
> > >
> > > | $\mu$ | 2 | 3 | 4 | 5 |
> > > |-----|-----|-----|-----|-----|
> > > | acc(%) | 0.72 | 0.88 | 0.92 | 0.92 |
> > >
> > > Based on the current training status of the model, we are concerned that it may not be possible to complete training on the real-world dataset for $\mu$ ablation by the end of 22nd/Nov. However, when $\mu=0$, it reduces to TDRL, whose result has been compared with Caring. Once further ablation study with $\mu$ in the middle is finished, we will update it ASAP.
> > >
> > > **Finally**, we sincerely appreciate your recognition of our efforts. We would also like to express our gratitude once again for the feedback you provided to help us improve the paper. Please do not hesitate to ask if you have further concerns. We wholeheartedly hope that we still have the opportunity to address all your concerns and doubts.

---

> > > ### Author Response · Authors · 2023-11-23
> > > **Could you please kindly let us know if our further response answers your questions?**
> > >
> > > Dear Reviewer 5cYn,
> > >
> > > Could you please kindly let us know if our further response answers your questions? Many thanks for your valuable suggestions and questions again! We hope that we can address all of your concerns!
> > >
> > > Best Wishes,
> > >
> > > Authors of submission 2088

---

> ### Author Response · Authors · 2023-11-23
> **Further explaination for abaltion study on real-world data**
>
> Dear Reviewer 5cYn,
>
> We managed to finish the ablation study on real-world data with different $\mu$, which is consistent with our intuition. Here is the result, and we will update it in our paper ASAP.
> |$\mu$|0|3|Full Length|
> |------|------|------|------|
> |acc(%)|38.58|40.67|41.22|
>
> Many thanks,
>
> Authors of submission 2088

---

### Official Review · Reviewer_WQAA · 2023-11-01

**Soundness:** 3 good
**Presentation:** 3 good
**Contribution:** 4 excellent
**Rating:** 6
**Confidence:** 3

**Summary:**

This paper addresses the problem of identifying the sources of a data-generating process (similar to ICA) where we assume a temporal scenario, and when the generator from the sources to the observation at a specific time is non-invertible. The authors then assume that the data-generating process is invertible, conditioned on sources from the past, and provide theoretical identifiability results up to permutations and component-wise transformations for three different scenarios. Then, a parametric approach based on variational auto-encoders and normalizing flows is proposed in order to learn the data-generating process, and put the test in synthetic and real-world experiments against previously-proposed approaches.

**Strengths:**

- The paper addresses an interesting variation of ICA with clear practical usage.
- The motivation of the paper is quite appealing.
- The theoretical results look quite impressive and of interest for the community (although I have not looked into the proofs in detail).
- Empirical results look in principle quite positive and validate the proposed architecture.

**Weaknesses:**

- The presentation of the paper leaves a lot to be desired.
  - W1. I don't fully understand why this work takes such a confrontational stance with respect to the ICA community. From my perspective, and please correct me if I am wrong, everything the paper does is taking the same ICA framework as [1] (non-linear ICA with z independent given another random variable), and assume that the generation process is invertible _given that same random variable_ (in this case, the previous sources). This is quite commendable and interesting, and complements the existing body of work, rather than being obfuscated on "non-invertibility" (which is not completely true).
  - W2. The manuscript does little effort in providing explanations and justifying certain statements (e.g. the entire paragraph before section 3).
  - W3. Similarly, the mathematical notation is far from standard, convoluted, sometimes wrong, and unnecessarily unwelcoming. E.g.:
    - In Eq. 3 $T \circ \pi \circ m$ should be in parentheses.
    - (I think that) the union symbol $\cup$ is used in places where the Cartesian product is meant to be (e.g. the continuity condition).
    - Conditions like those from Eq. 6 are overly convoluted for no reason as, e.g., $v_{k,t}$ being linearly independent could be much simplified by saying that the Hessian has non-zero determinant (i.e. is invertible).
    - Jargon is non-consistent, e.g., secondary differentiable, second-order differentiable, and second order differentiable. Similarly, normalizing flows are then called normalized flows.
  - W4. I also find section 4 a bit too convoluted to read, and it takes several reads to understand how exactly looks the network proposed by the authors. My advice would be to try to make more explicit the connections between each network component and the theory/data-generating process.

About the experiments:
- W5. I am surprised that there are no comparisons with iVAE, despite being cited.
- W6. Number of parameters as well as training times for the real-world experiments seem necessary to me.
- W7. While real-world results are ok, I find the discussion deceivingly positive, since CMCIR obtains better results on average and beats CaRiNG quite significantly in some individual question types.


[1] Khemakhem, I., Kingma, D., Monti, R., & Hyvarinen, A. (2020, June). Variational autoencoders and nonlinear ICA: A unifying framework. In International Conference on Artificial Intelligence and Statistics (pp. 2207-2217). PMLR.

**Questions:**

- Q1. I don't think I understand what is the column "All" in Table 2. Is it the mean of the other columns? Because if that is the case, these numbers do not add up.

---

> ### Author Response · Authors · 2023-11-18
> **Response to Reviewer WQAA Part 1**
>
> Dear Reviewer WQAA, we greatly appreciate your time dedicated to reviewing this paper and all your effort to improve its writing quality.
> We have rewritten the manuscript and appendix to make it as clear as possible and provided the point-to-point responses.
>
> > **W1:** I don't fully understand why this work takes such a confrontational stance with respect to the ICA community. From my perspective, and please correct me if I am wrong, everything the paper does is taking the same ICA framework as [1] (non-linear ICA with z independent given another random variable), and assume that the generation process is invertible given that same random variable (in this case, the previous sources). This is quite commendable and interesting, and complements the existing body of work, rather than being obfuscated on "non-invertibility" (which is not completely true).
>
> **A1:** We highly appreciate your insightful suggestions which have helped clarify our paper's position and contribution. Besides, we are also grateful that you think our method is commendable and interesting.  In response to your comments, we have made significant modifications to the introduction of our revised paper, ensuring a more coherent and accurate representation of our work. Below are the key aspects of these modifications:
>
> **Clarifying non-confrontational stance** We apologize for the misleading presentation of the confrontational stance. Actually, our work is not intended to be confrontational towards the ICA community. Instead, our aim is to build upon and extend the existing body of work.
>
> **Affirmation of following the ICA framework.** We confirm that our paper is firmly rooted in the nonlinear ICA framework (thank you for pointing it out). Our contribution is in extending this framework to scenarios where temporal dynamics involve a non-invertible mixing function.  This extension implies that the current observation $x\underline{ }t$ does not fully include all the information about the latent variable $z\underline{ }t$. To make our contribution clear, we have enriched the introduction with a comprehensive overview of the current nonlinear ICA literature. This also serves to better highlight our method's unique contributions and its role in complementing and enriching the existing body of work.
>
> We hope that these modifications adequately address your concerns and better articulate the intent and scope of our research.

---

> ### Author Response · Authors · 2023-11-18
> **Response to Reviewer WQAA Part 2**
>
> > **W2:** The manuscript does little effort in providing explanations and justifying certain statements (e.g. the entire paragraph before section 3).
>
> **A2:** Thank you very much for your suggestions on improving the readability of our paper. We have revised the manuscript accordingly. Please kindly refer to our revised paper for the details. Please let us know if you have any concerns about the provided explanations and justification.
>
> **Notice:** Due to the issue of markdown for equations, we use $a\underline{ }(b)$ to represent $a_b$.
>
> **Modification of Section 2.2** At the end of Section 2.2, we clarified the differences between our Definition 1 and the identifiability definition in existing literature.
> Different from the existing literature, we involve $\mathbf{m}$ in the above definition, since it does so implicitly as a property of the mixing function $\mathbf{g}$, although it does not explicitly participate in the generation process. Furthermore, the identifiability of $\mathbf{g}$ is different. In previous nonlinear ICA methods, such as TDRL and iVAE, the mixing function $\mathbf{g}$ is identifiable.
> However, in our case, we cannot find the identifiable mixing function since the information loss is caused by non-invertibility. Instead, we can obtain a component-wise transformation of a permuted version of latent variables $\mathbf{\hat{z}}\underline{ }t = \mathbf{m}(\mathbf{x}\underline{ }{t:t-\mu})$.
> The latent causal relations are also identifiable, up to a permutation $\pi$ and component-wise invertible transformation $T$, i.e., $\mathbf{\hat{f}} = T \circ \pi \circ \mathbf{f}$, once $\mathbf{z}\underline{ }t$ is identifiable.
> It is because, in the time-delayed causally sufficient system, the conditional independence relations fully characterize time-delayed causal relations when we assume no latent causal confounders in the (latent) causal processes.
>
>
> **Further illustration with intuitive examples**
> Consider a rapidly moving ball on a two-dimensional plane as described in Figure 2. The horizontal and vertical coordinates of the ball's position at any given moment can be represented by the latent variable $\mathbf{z}\underline{ }t\in \mathbb{R}^2$. We assume that the ball follows a curved trajectory constrained by the nonlinear function $\mathbf{f}$  as it moves.
>
> Suppose that we observe the ball with a visual persistence effect, where each observation $\mathbf{x}\underline{ }t$ captures several consecutive latent variables as
> $\mathbf{x}\underline{ }t = \mathbf{g}(\mathbf{z}_{<t})$.
> The mixing function $\mathbf{g}$ refers to the weighted sum of the images obtained through multiple exposures, which is what a person ultimately observes as $\mathbf{x}\underline{ }t$. In this case, the invertibility of the mapping from $\mathbf{z}\underline{ }{t}$ to $\mathbf{x}\underline{ }{t}$ is compromised since the current frame also contains the latent information from previous frames.
>
> **Further illustration with mathematical examples**
> We also provided a mathematical example to demonstrate the existence of function $\mathbf{m}$. Following the concept of visual persistence, let the current observation be a weakened previous observation overlaid with the current image of the object, i.e., $\mathbf{x}\underline{ }t = \mathbf{z}\underline{ }t + \frac{1}{2}\mathbf{x}\underline{ }{(t-1)} = \sum_{i=1}^{\infty} \left(\frac{1}{2}\right)^{i} \mathbf{z}\underline{ }{(t-i)}$.  Given an extra observation, the current latent variable can be rewritten as $\mathbf{z}\underline{ }t  = \mathbf{x}\underline{ }t - \frac{1}{2}\mathbf{x}\underline{ }{(t-1)}$.
> Thereby we can easily recover latent variables that cannot be obtained from a single observation, i.e., $\mathbf{z}\underline{ }t = \mathbf{m}(\mathbf{x}\underline{ }{(t:t-1)}) = \mathbf{x}\underline{ }t - \frac{1}{2}\mathbf{x}\underline{ }{(t-1)}$.

---

> ### Author Response · Authors · 2023-11-18
> **Response to Reviewer WQAA Part 3**
>
> > **W3:** Similarly, the mathematical notation is far from standard, convoluted, sometimes wrong, and unnecessarily unwelcoming. E.g.:
>
> >1. In Eq. 3 $T\circ\pi\circ m$ should be in parentheses;
>
> >2. (I think that) the union symbol is used in places where the Cartesian product is meant to be (e.g. the continuity condition)
>
> >3. conditions like those from Eq. 6 are overly convoluted for no reason as, e.g., being linearly independent could be much simplified by saying that the Hessian has non-zero determinant (i.e. is invertible).
>
> >4. Jargon is non-consistent, e.g., secondary differentiable, second-order differentiable, and second order differentiable. Similarly, normalizing flows are then called normalized flows.
>
> **A3:** We sincerely appreciate for pointing out the mathematical notation errors and reader-unfriendly writing in our article. As a result, we have made the following modifications:
> - we put $T\circ\pi\circ m$ in parentheses in Eq. 3;
> - we rewrite Lemma 2 and change the domain of $h$ to $\mathcal{\hat{Z}}\times\mathcal{C}$, which refers to the Cartesian product;
> - we add an explanation of the sufficiency assumption before the proof of Theorem 1, providing its matrix form in Eqs.12,13 and meaning to make it easier to understand. Please kindly note that $\mathbb{V}\underline{ }t$ and $\mathbb{\mathring{V}}\underline{ }t$ are not Hessian matrices, but rather mixed partial derivatives of order 2 and 3;
> - we review the consistency and accuracy of the terminology throughout the entire document and have made consistent changes to "second-order" and "normalizing flow".
>
> After all, thank you for your efforts in helping us improve the quality of our presentation.
>
> > **W4:** I also find section 4 a bit too convoluted to read, and it takes several reads to understand how exactly looks the network proposed by the authors. My advice would be to try to make more explicit the connections between each network component and the theory/data-generating process.
>
>
> **A4:** We are grateful for your valuable advice which helped make the implementation clearer. In light of your advice, we reorganized Section 4 and highlighted the connections between network components and the assumptions in our theorem. Please kindly refer to the revised paper and let us know if you have any further suggestions.
>
> > **W5:** I am surprised that there are no comparisons with iVAE, despite being cited.
>
> **A5:** Thanks for pointing it out.
> The reason why we did not compare with iVAE is that, theoretically, iVAE requires additional domain information to achieve identifiability, which is not valid in our setting with stationary transitions and no domain information. We apply this stationary setting for the comparison since it is somehow more difficult than the non-stationary one, since no extra domain information is used.  Given these considerations, we only compare methods that work in stationary scenarios.
>
> We agree that iVAE is an important work in the literature of ICA, and it is valuable to compare with it. We have included the comparative results in Table 1 in our revised paper, made modifications to Figure 4 (originally Figure 3), and provided relevant explanations in the "Baseline Methods" paragraph of Section 5.1.
> Please note that the input of iVAE requires a domain label. We simply use the time index as the domain label.
>
> > **W6:** Number of parameters as well as training times for the real-world experiments seem necessary to me.
>
> **A6:** We kindly appreciate your suggestions and we provide these details in the following table. Compared with HCRN, we had a similar number of parameters, and needed double the training time.  It is because we apply the normalizing flow to calculate the Jacobian matrix, which introduces the extra computation cost. Please kindly note that, despite more training time, the inference efficiency is totally the same, since the normalizing flow is not used for inference (only for calculating KL loss).  We have included this analysis in Appendix A4.1 and Table A3.
>
> | Method                  | HCRN     | CaRiNG   |
> | ----------------------- | -------- | -------- |
> | Number of Parameters    | 42278786 | 43721954 |
> | Training Time per Epoch | 6min54s  | 13min26s |
> | Inference Time per Epoch| 49s/epoch| 49s/epoch|

---

> ### Author Response · Authors · 2023-11-18
> **Response to Reviewer WQAA Part 4**
>
> > **W7:**  While real-world results are ok, I find the discussion deceivingly positive, since CMCIR obtains better results on average and beats CaRiNG quite significantly in some individual question types.
>
> **A7:** Thanks for your questions, which help us clarify the different settings with CMCIR.
>
> We would like to highlight that the experimental setting of CMCIR is a little different from the other baselines.  Specifically, all the other baselines and our method apply a pre-trained ResNet-101 model [R1] as the fine-grained frame feature, and a pre-trained MobileNetv2 [R2] is used as the lightweight CNN to extract.  As mentioned in  CMCIR,
> "The Swin-L [R3] pre-trained on the ImageNet-22K dataset is used to extract the frame-level appearance features, and the video SwinB [R4] pre-trained on Kinetics-600 is applied to extract the clip-level motion features". Please note that Swin-L and SwinB are much stronger backbone than ResNet we used.
>
>
>
> > **W8(Q1):** I don't think I understand what is the column "All" in Table 2. Is it the mean of the other columns? Because if that is the case, these numbers do not add up.
>
> **A8:** Thanks a lot for your question which enhanced our readability. The column ''ALL'' denotes the weighted average of single-item scores, with the weight being the proportion of this type of question in the test set. We have added an explanation in the caption of Table 2 accordingly.
>
> [R1] He K, Zhang X, Ren S, et al. Deep residual learning for image recognition[C]//Proceedings of the IEEE conference on computer vision and pattern recognition. 2016: 770-778.
>
> [R2] Sandler M, Howard A, Zhu M, et al. Mobilenetv2: Inverted residuals and linear bottlenecks[C]//Proceedings of the IEEE conference on computer vision and pattern recognition. 2018: 4510-4520.
>
> [R3] Liu Z, Lin Y, Cao Y, et al. Swin transformer: Hierarchical vision transformer using shifted windows[C]//Proceedings of the IEEE/CVF international conference on computer vision. 2021: 10012-10022.
>
> [R4] Liu Z, Ning J, Cao Y, et al. Video swin transformer[C]//Proceedings of the IEEE/CVF conference on computer vision and pattern recognition. 2022: 3202-3211.

---

> > ### Comment · Reviewer_WQAA · 2023-11-20
> >
> > Dear authors,
> >
> > Thanks a lot for the extensive changes to the manuscript. I believe that it does indeed improve the readability of the manuscript quite significantly, and the additional examples help gain some intuition on the problem at hand.
> >
> > I have an additional question: When you say that "$\mathbb{V}_t$ is not a Hessian matrix, but rather mixed partial derivatives of order 2;" Am I getting it wrong, or isn't that precisely a flattened version of the Hessian matrix wrt z, with extra zeroes at the end?
> >
> > Just as a sidenote, I will decide if I update my score after the end of the rebuttal period.

---

> > > ### Author Response · Authors · 2023-11-20
> > > **Further Response on the Difference Between our Sufficiency Assumption and Hessian matrix**
> > >
> > > Dear reviewer WQAA,
> > >
> > > Thank you for your quick response and affirmation of the modifications we made. Regarding the new question, we have $\mathbb{V}_t = $
> > >
> > > $$
> > >  \begin{bmatrix}
> > >         \frac{\partial^2 \eta_{1t}}{\partial z_{1,t} \partial z_{1,t-r-1}} &
> > >         \frac{\partial^2 \eta_{1t}}{\partial z_{1,t} \partial z_{2,t-r-1}} &
> > >         \cdots &
> > >         \frac{\partial^2 \eta_{1t}}{\partial z_{1,t} \partial z_{n,t-r-1}} \newline
> > >         \frac{\partial^2 \eta_{2t}}{\partial z_{2,t} \partial z_{1,t-r-1}} &
> > >         \frac{\partial^2 \eta_{2t}}{\partial z_{2,t} \partial z_{2,t-r-1}} &
> > >         \cdots &
> > >         \frac{\partial^2 \eta_{2t}}{\partial z_{2,t} \partial z_{n,t-r-1}} \newline
> > >         \vdots & \vdots & \ddots & \vdots \newline
> > >         \frac{\partial^2 \eta_{nt}}{\partial z_{n,t} \partial z_{1,t-r-1}} &
> > >         \frac{\partial^2 \eta_{nt}}{\partial z_{n,t} \partial z_{2,t-r-1}} &
> > >         \cdots &
> > >         \frac{\partial^2 \eta_{nt}}{\partial z_{n,t} \partial z_{n,t-r-1}}
> > >     \end{bmatrix}^T
> > > $$
> > > as a second-order mixed derivative of a n-dimension vector $[\eta_{1t}, \eta_{2t}, \cdots, \eta_{nt}]$, where each is $\eta_{kt}: \mathbb{R}^{n\times r}\rightarrow\mathbb{R}$. However, Hessian matrix is defined on $f:\mathbb{R}^n\rightarrow\mathbb{R}$, i.e.,
> > > $$
> > > \mathbb{H}_f = \begin{bmatrix}
> > >         \frac{\partial^2 f}{\partial x_1 \partial x_1} &
> > >         \frac{\partial^2 f}{\partial x_1 \partial x_2} &
> > >         \cdots &
> > >         \frac{\partial^2 f}{\partial x_1 \partial x_n} \newline
> > >         \frac{\partial^2 f}{\partial x_2 \partial x_1} &
> > >         \frac{\partial^2 f}{\partial x_2 \partial x_2} &
> > >         \cdots &
> > >         \frac{\partial^2 f}{\partial x_2 \partial x_n} \newline
> > >         \vdots & \vdots & \ddots & \vdots \newline
> > >         \frac{\partial^2 f}{\partial x_n \partial x_1} &
> > >         \frac{\partial^2 f}{\partial x_n \partial x_2} &
> > >         \cdots &
> > >         \frac{\partial^2 f}{\partial x_n \partial x_n}
> > > \end{bmatrix}^T,
> > > $$
> > >  which is different from our setting. They are different in form, but as you mentioned, they both utilize information from second-order derivatives. We attempt to establish a connection between them, but it is challenging to formalize them consistently because one is at the vector level and the other is the derivative at a point. Please kindly let us know if you have any further concerns. We are eager and happy to discuss with you.

---

> > > ### Author Response · Authors · 2023-11-21
> > > **Does our new response address your concerns?**
> > >
> > > Dear reviewer WQAA,
> > >
> > > Does our new response address your concerns? If you have any further questions, please feel free to ask us at any time.

---

> > > > ### Comment · Reviewer_WQAA · 2023-11-21
> > > >
> > > > Dear authors,
> > > >
> > > > Yes, they did clarify it. For me, what you wrote as $\mathbb{V}_t$ I also call it a Hessian (of $\eta$ wrt. $z$), but now I understand where was the misunderstanding coming from.
> > > >
> > > > Thanks again for the detailed responses!

---

> > > > > ### Author Response · Authors · 2023-11-21
> > > > > **Thanks for your feedback.**
> > > > >
> > > > > Dear Reviewer WQAA,
> > > > >
> > > > > We highly appreciate your swift reply and engaging discussion. We are glad to hear that your concerns have been clarified. We look forward to possibly receiving an improved score at the end of the rebuttal period.
> > > > > Thanks for all your effort and time on this paper again!
> > > > >
> > > > > Best Regards,
> > > > > Authors of Submission 2088

---

> > > > > ### Author Response · Authors · 2023-11-23
> > > > >
> > > > > Dear reviewer WQAA,
> > > > >
> > > > > Thanks for your feedback and suggestions to improve our paper. Given that it is at the end of the rebuttal phase,  and all your concerns have been well resolved. Could you please raise the scores to reflect it? It is very important to us. Thanks again for all of your efforts and time dedicated to this paper.
> > > > >
> > > > > Best Regards,
> > > > >
> > > > > Authors of submission 2088

---

> > > > > > ### Comment · Reviewer_WQAA · 2023-11-23
> > > > > >
> > > > > > Dear authors, I have updated my score to reflect my view after the rebuttal period. Thanks again for the fruitful discussion!

---

### Meta-Review · Area_Chair_nTwL · 2023-12-05

**Metareview:**

This paper has a very ambitious goal in building identifiable deep generative models without using invertible generative/mixing processes. The main idea is to use temporal information (e.g., consecutive frames) to recover the latent variables. The paper's contribution is mainly theoretical, although a corresponding deep generative model is proposed with some experimental validations.

While reviewers all agree that the paper's contribution could be very significant if the results are sound, reviewer ur5z raised questions regarding the practicality of the assumptions in practice as well as substantial number of confusing theoretical arguments. Other reviewers also raised questions regarding experimental results and the novelty of the constructed deep generative models.

The authors incorporated many suggestions from the reviewers, making the revised paper significantly different from the first submission. In particular, as reviewers ur5z and uMXb noted, the generative model has been changed to include the previous z-values, which changes both the theoretical results as well as the implementation significantly. All the reviewers agree that the revised version contains very significant difference that requires another round of full review.

**Justification For Why Not Higher Score:**

The first version of the paper contains theoretical issues, which led to significant amount of revisions that I believe warrant another round of full review.

Reference according to AC guide: https://iclr.cc/Conferences/2024/ACGuide

Q: How do the reviewers and ACs deal with the revisions of the paper during the discussion period?

A: The authors may revise their submission up until November 18, one week before the end of the discussion period. The final decision and meta-review should take the revisions into account; **however, ACs reserve the right to ignore this revision if it is substantially different from the original version.**

**Justification For Why Not Lower Score:**

N/A

---

### Decision · Program_Chairs · 2024-01-16

Reject